# Off-Policy Safe Reinforcement Learning with Constrained Optimistic Exploration

**Guopeng Li**
Faculty of Mechanical Engineering
Delft University of Technology
Delft, the Netherlands
g.li-5@tudelft.nl

**Matthijs T. J. Spaan**
Faculty of Electrical Engineering,
Mathematics and Computer Science
Delft University of Technology
Delft, the Netherlands
m.t.j.spaan@tudelft.nl

**Julian F. P. Kooij**
Faculty of Mechanical Engineering
Delft University of Technology
Delft, the Netherlands
j.f.p.kooij@tudelft.nl

## Abstract

When safety is formulated as a limit of cumulative cost, safe reinforcement learning (RL) aims to learn policies that maximize return subject to the cost constraint in data collection and deployment. Off-policy safe RL methods, although offering high sample efficiency, suffer from constraint violations due to cost-agnostic exploration and estimation bias in cumulative cost. To address this issue, we propose Constrained Optimistic eXploration Q-learning (COX-Q), an off-policy safe RL algorithm that integrates cost-bounded online exploration and conservative offline distributional value learning. First, we introduce a novel cost-constrained optimistic exploration strategy that resolves gradient conflicts between reward and cost in the action space and adaptively adjusts the trust region to control the training cost. Second, we adopt truncated quantile critics to stabilize the cost value learning. Quantile critics also quantify epistemic uncertainty to guide exploration. Experiments on safe velocity, safe navigation, and autonomous driving tasks demonstrate that COX-Q achieves high sample efficiency, competitive test safety performance, and controlled data collection cost. The results highlight COX-Q as a promising RL method for safety-critical applications.

## 1 Introduction

Many real-world decision-making tasks have safety requirements. For example, robots must not harm humans (Luo et al., 2025), and autonomous vehicles must avoid collisions (Feng et al., 2023). Such concerns motivate *safe reinforcement learning (RL)*, which commonly formulates the problem as a constrained Markov decision process (CMDP) (Altman, 2021). In such a setting, the agent aims to maximize the return while keeping the cumulative safety cost below a threshold. The growing interest in the deployment of RL has led to increased attention to safe RL (Brunke et al., 2022).

Collecting data directly from the environment is imperative for many RL applications due to the limited fidelity of simulation or the need for human-in-the-loop interactions. For example, in autonomous driving in mixed traffic (Chen et al., 2024) and healthcare advising (Gottesman et al., 2019) tasks, agents must collect data safely in the real world. Therefore, *sample efficiency* is critical for safe RL, as it directly determines the cost of data collection.

Off-policy RL has higher sample efficiency than on-policy methods through experience replay (Chen et al., 2021) and active exploration (Ladosz et al., 2022). However, applying off-policy methods to safe RL faces substantial challenges. First, the underestimation bias in the cumulative cost often leads to unsafe policies (Wu et al., 2024). Second, exploration in off-policy RL lacks cost constraints. The agent can be misled into risky areas, causing uncontrolled data collection costs.

Therefore, existing safe RL methods are predominantly on-policy (Gu et al., 2024b). Off-policy approaches struggle to satisfy cost constraints in data collection and deployment, as shown in the OmniSafe benchmark (Ji et al., 2024). These issues highlight a critical knowledge gap:

*How can off-policy safe RL maintain high data efficiency and meanwhile achieve robust constraint satisfaction in both data collection and deployment, through cost-constrained exploration and reliable value learning?*

To address this challenge, we propose *Constrained Optimistic eXploration Q-learning (COX-Q)*, an off-policy primal-dual safe RL algorithm. COX-Q integrates a novel cost-bounded optimistic exploration strategy with conservative value learning based on mixed quantile critics. COX-Q demonstrates competitive performance on various safe RL benchmarks, showcasing its effectiveness for safety-critical applications.

## 2 RELATED WORK

This section provides a concise overview of related work to contextualize the core contributions of this study. We first clarify some key terminologies and define the scope of the overview. Safe RL is a broad concept that involves a wide range of methodologies, such as Control Barrier Functions (CBFs) (Chen et al., 2024), reachability methods (Ganai et al., 2023). We focus on the formulation of safety as constraints on cumulative costs, and address it within the constrained RL framework (Altman, 2021). Additionally, this overview comprises only model-free safe RL methods. Model-based methods (e.g., Safe Dreamer (Huang et al., 2023)) are not included due to fundamental differences. Related methods are grouped into on-policy and off-policy categories.

Most existing safe RL methods are on-policy, as sharing the behaviour and target policies allows each update to directly enforce constraint satisfaction through adjusted gradients or trust region techniques. On-policy approaches include first-order methods such as FOCOPS (Zhang et al., 2020) and CUP (Yang et al., 2022), as well as second-order methods like CPO (Achiam et al., 2017), and RCPO (Tessler et al., 2018). Other variants include the PID-Lagrangian method (Stooke et al., 2020), risk-aware scheduling methods such as Saute RL (Sootla et al., 2022a) and PPOSimmer (Sootla et al., 2022b), and the early terminated MDP formulation (Sun et al., 2021). These methods and their variants have demonstrated strong empirical performance in many safe RL benchmarks. For a comprehensive review, we refer readers to (Gu et al., 2024b).

In contrast, off-policy safe RL is less studied. Most approaches adopt primal-dual methods like Lagrangian and PID-Lagrangian (Stooke et al., 2020), but suffer from poor safety performance due to the underestimation bias in cost values, often leading to constraint violations. To mitigate this, conservative cost estimators have been proposed. For example, Worst-Case SAC (WCSAC) (Yang et al., 2021) penalizes underestimated costs to improve constraint satisfaction. CAL (Wu et al., 2024) further accelerates training using local policy convexification and the augmented Lagrangian method, achieving strong safety and sample efficiency using a high update-to-data (UTD) ratio. In terms of exploration, Gao et al. (2025) proposed the so-called MICE to address the underestimation of cost. The key idea is to use a memory-based intrinsic cost around unsafe states so the cost critic conservatively overestimates risk. Although the original implementation is for on-policy methods, in principle, the idea can be adopted to off-policy approaches. A recent study by McCarthy et al. (2025) incorporates optimistic actor-critic (OAC) (Ciosek et al., 2019) into off-policy safe RL. The resulting ORAC algorithm actively explores regions with potentially higher reward and lower cost. While ORAC shows robust safety performance in tests, as the authors state, it does not enforce cost constraints in data collection. How to realize cost-compliant exploration remains an open challenge.

In summary, a key gap in off-policy safe RL is the lack of a principled cost-constrained exploration strategy integrated with conservative value learning. Our approach addresses this challenge from both theoretical and practical aspects.

## 3 PROBLEM FORMULATION

Consider a CMDP defined by $(S, A, r, c, p, p_0, \gamma, d)$. $S \subseteq \mathbb{R}^m$ is the state space. For a state $s_t \in S$, an agent controlled by a policy $a \sim \pi(\cdot|s)$ takes an action $a_t$ in the action space $A \subseteq \mathbb{R}^n$, then the next state follows $p(s_{t+1}|s_t, a_t)$. The agent receives a reward $r_t \in \mathbb{R}$ and pays a non-negative cost

$c_t \in \mathbb{R}^+$. The distribution of the initial state is $p_0(s_0)$. $\gamma \in (0,1)$ is the discount factor shared by the cumulative reward $Z_r^\pi$ and cost $Z_c^\pi$, which are both *random variables*:

$$Z_r^\pi(s_t, a_t) = \sum_{k=0}^{\infty} \gamma^k r_{t+k+1}, \quad Z_c^\pi(s_t, a_t) = \sum_{k=0}^{\infty} \gamma^k c_{t+k+1}. \tag{1}$$

The state-action value functions (Q-functions) capture the expected return and cost for the policy:

$$Q_r^\pi(s_t, a_t) = \mathbb{E}_\pi[Z_r^\pi(s_t, a_t)], \quad Q_c^\pi(s_t, a_t) = \mathbb{E}_\pi[Z_c^\pi(s_t, a_t)]. \tag{2}$$

In this setting, safe RL considers a constrained optimization problem:

$$\max_\pi \mathbb{E}_{s \sim \rho_\pi, a \sim \pi(\cdot|s)}[Q_r^\pi(s, a)], \quad \text{s.t.} \quad \mathbb{E}_{s \sim \rho_\pi, a \sim \pi(\cdot|s)}[Q_c^\pi(s, a)] \le d, \tag{3}$$

where $\rho_\pi$ is the state density function of $\pi$, and $d$ is a cost threshold that should not be exceeded to ensure safety. The primal-dual approach constructs the following dual form, updating the policy $\pi$ and Lagrangian multiplier $\lambda$ iteratively:

$$\min \mathbb{E}_{s \sim \rho_\pi, a \sim \pi(\cdot|s)}[Q_r^\pi(s, a) - \lambda(Q_c^\pi(s, a) - d)], \tag{4}$$

$$\arg \min_{\lambda > 0} \lambda \times (d - \mathbb{E}_{s \sim \rho_\pi, a \sim \pi(\cdot|s)}[Q_c^\pi(s, a)]). \tag{5}$$

In summary, for safe RL, we have two factors that impact exploration and policy learning:

- A cost limit $d$ divides $(s, a)$ into safe ($Q_c^\pi \le d$) and unsafe ($Q_c^\pi > d$) regions.
- Two objectives, $Q_r^\pi(s, a)$ for the return and $Q_c^\pi(s, a)$ for the cumulative cost.

It is also useful to note that $d$ is the cost limit for both data collection (training) and tests. This requirement is naturally satisfied for on-policy methods, but *not* for off-policy methods. Next, we introduce the proposed COX-Q algorithm in detail.

## 4 COST-CONSTRAINED OPTIMISTIC EXPLORATION

Off-policy RL for continuous control tasks can use Optimistic Actor-Critic (OAC) (Ciosek et al., 2019) for active exploration. In single-objective RL, OAC first estimates an optimistic upper bound of Q-value $\hat{Q}^{\text{UB}}(s, a)$ from an ensemble of critics, then maximizes this objective under a KL divergence constraint (trust region). If the target policy is $\mathcal{N}(\mu_T, \Sigma_T)$, then the OAC exploration policy $\mathcal{N}(\mu_E, \Sigma_E)$ is given by the following proposition in the original paper (Ciosek et al., 2019):

$$\mu_E = \mu_T + \sqrt{2\delta} \times \frac{\Sigma_T [\nabla_a \hat{Q}^{\text{UB}}(s, a)]_{a=\mu_T}}{\left\| [\nabla_a \hat{Q}^{\text{UB}}(s, a)]_{a=\mu_T} \right\|_{\Sigma_T}}, \quad \Sigma_E = \Sigma_T, \tag{6}$$

where $\delta$ is a threshold value for the KL-divergence between the target and the exploration policies, which is a hyperparameter. For convenience, we can rewrite the displacement $\mu_\Delta$ of the mean action in equation 6 in terms of the total gradient $g_t$ and the step length $\eta$ as follows:

$$\mu_\Delta = \mu_E - \mu_T = \eta \Sigma_T g_t, \quad \text{where} \quad \eta = \sqrt{\frac{2\delta}{g_t^\mathsf{T} \Sigma_T g_t}}, \quad g_t = \nabla_a \hat{Q}^{\text{UB}}(s, a)|_{a=\mu_T} \tag{7}$$

For safe RL, the exploration policy is expected to fully explore safe regions, keep the number of visits to unsafe regions below the cost limit, and prevent any objective (return or cost) from dominating the exploration. Therefore, we propose the Cost-Constrained Optimistic eXploration (COX) strategy, which extends the single-objective OAC (Ciosek et al., 2019) to multi-objective safe RL settings. In principle, COX exploration sequentially determines *(1) the effective exploration direction $g^*$* and *(2) the safe exploration step length $\eta^*$*, to replace $g_t$ and $\eta$ in equation 7, respectively. All theories in this section are based on the assumption of Gaussian policies, which are compatible with most mainstream off-policy RL methods, such as Soft Actor-Critic (SAC) (Haarnoja et al., 2018).

### 4.1 POLICY-MGDA FOR EXPLORATION GRADIENT CONFLICT RESOLUTION

Since safe RL involves two objectives, we first determine the aligned, non-conflicting exploration direction $g^*$ in terms of these objective gradients. Omitting superscript $\pi$, we denote:

$$g_r = \nabla_a \hat{Q}_r^{\text{UB}}(s,a)|_{a=\mu_T} \quad g_c = \nabla_a \hat{Q}_c^{\text{LB}}(s,a)|_{a=\mu_T}, \quad g_m = \nabla_a \hat{Q}_c^{\text{mean}}(s,a)|_{a=\mu_T}, \quad (8)$$

where superscripts "UB" and "LB" represent estimated optimistic upper and lower bound, respectively. Note that the dual form in equation 4 favours higher reward and lower cost.

**In safe regions:** Within safe regions ($Q_c^\pi(s,a) \leq d$), the KKT condition of equation 3 indicates that the constraint is not activated. The exploration considers the return along, thus $g^* = g_r$.

**In unsafe regions:** Within unsafe regions, the gradient for the overall objective according to equation 4 is $g_r - \lambda g_c$. However, this naive sum should not be used directly for $g^*$ since we further want to ensure that both return and cost are improving:

$$\Delta \hat{Q}_c^{\text{LB}}(s,\mu_E) = g_c^\mathsf{T}\mu_\Delta = \eta \times g_c^\mathsf{T}\Sigma_T g_t \leq 0 \quad \text{and} \quad \mu_\Delta \hat{Q}_r^{\text{UB}}(s,\mu_E) = g_r^\mathsf{T}\mu_\Delta = \eta \times g_r^\mathsf{T}\Sigma_T g_t \geq 0. \quad (9)$$

If one of the conditions in equation 9 is violated, we say that the *exploration gradients conflict*. For example, if $g_r$ dominates the exploration in unsafe regions, then the agent may be misled deeper towards the unsafe side. Note that $\eta$ is non-negative, so whether gradients conflict and the magnitude of the conflict is measured by $\Sigma$-*metric*:

$$\langle g_i, g_j \rangle_{\Sigma_T} \equiv g_i^\mathsf{T}\Sigma_T g_j, \quad (10)$$

This metric is in the action space, so the covariance matrix of the policy is included. This is different from multi-task learning that uses the direct inner product in the model parameter space (Zhang & Yang, 2021). To resolve exploration gradient conflicts, we extend the Multiple Gradient Descent Algorithm (MGDA) (Désidéri, 2012) to the action space, forming the so-called *Policy-MGDA*. We first define a gradient space (a "hyper-cone") in which both conditions of equation 9 hold:

$$K := \{g : v_r = \langle g_r, g \rangle_{\Sigma_T} \geq 0, v_c = \langle -g_c, g \rangle_{\Sigma_T} \geq 0\}. \quad (11)$$

For two gradient vectors, such a $K$ always exists except for degraded or co-linear cases. Then we find the optimal $g^*$ that best aligns with the original direction $g_r - \lambda g_c$ w.r.t $\Sigma$-metric:

$$g^* = \arg\min_{u \in K} \|u - (g_r - \lambda g_c)\|_{\Sigma_T}^2. \quad (12)$$

**Lemma 1** *We denote $g_{raw} = g_r - \lambda g_c$ and the following Gram-scalars and multipliers:*

$$s_{ij} = \langle g_i, g_j \rangle_{\Sigma_T}, \quad v_i = \langle g_t, g_i \rangle_{\Sigma_T}, \quad \mu_r = \frac{-s_{cc}v_r + s_{rc}v_c}{s_{rr}s_{cc} - s_{rc}^2}, \quad \mu_c = \frac{-s_{rc}v_r + s_{rr}v_c}{s_{rr}s_{cc} - s_{rc}^2} \quad (13)$$

*Then the optimal solution for equation 12 is:*

$$g^* = \begin{cases} g_{raw} & \text{if } g_t \in K \\ g_{raw} - \dfrac{v_r}{s_{rr}}g_r & \text{if } v_r < 0 \text{ and } v_c \leq 0 \\ g_{raw} - \dfrac{v_c}{s_{cc}}g_c & \text{if } v_r \geq 0 \text{ and } v_c > 0 \\ g_{raw} - \mu_r g_r + \mu_c g_c & \text{if } v_r < 0 \text{ and } v_c > 0 \end{cases} \quad (14)$$

The proof is in Appendix A.1. $g^*$ is the aligned exploration direction in unsafe regions. Note that policy-MGDA operates in the action space during the *online* data collection stage, which makes it fundamentally different from existing gradient manipulation methods operating in the *offline* model update stage (Gu et al., 2024a; Chow et al., 2021; Liu et al., 2022).

### 4.2 ADAPTIVE STEP LENGTH FOR EXPLORATION COST CONTROL

Given the exploration gradient $g^*$, we now determine the step length $\eta^*$, which controls the data collection cost. Both the microscopic single-step exploration and the macroscopic training progress are considered.

For each exploration step, the original single-objective OAC does not involve the cost constraint in equation 3. To address this issue, we explicitly bound the cost expectation by adjusting the step length $\eta$. Given the exploration direction $g^*$, the threshold of non-negative violation along this direction is the hinge:

$$\phi(\eta) = [\Delta \hat{Q}_c^{\text{mean}} - (d - \hat{Q}_c^{\text{mean}})]_+ = [\eta \langle g_m, g^* \rangle_{\Sigma_T} - (d - \hat{Q}_c^{\text{mean}})]_+. \tag{15}$$

Then we can formulate the following bi-level optimization problem:

$$\arg \max_{\eta^*} \eta^* \quad \text{s.t.} \quad 0 \le \eta^* \le \eta, \quad \phi(\eta^*) = \min_{0 \le \xi \le \eta} \phi(\xi). \tag{16}$$

This means that, once the full exploration step length makes the mean cost exceed $d$, we choose the maximum $\eta^*$ in the trust region to ensure the cost constraint violation $\phi(\eta)$ is 0 or minimized.

**Lemma 2** $g_m$ *is defined in equation 8. We further denote:*

$$s = \langle g_m, g^* \rangle_{\Sigma_T}, \quad r = d - \hat{Q}_c^{mean} \tag{17}$$

*Then the optimal solution for equation 16 is:*

$$\eta^* = \begin{cases} \eta & \text{if } s < 0 \\ 0 & \text{if } s > 0 \text{ and } r < 0; \text{ or } s = 0 \\ \min(\eta, r/s) & \text{if } s > 0 \text{ and } r \ge 0 \end{cases} \tag{18}$$

The proof is given in Appendix A.2. Nevertheless, equation 18 is not always valid. When $g^*$ tends to 0 around the optimum, $s \to 0$. So, the oscillating sign of $g^*$ makes $\eta^*$ jump between $\pm\eta$, manifesting as a pure extra action noise. To address this issue, we further adaptively adjust $\delta$, thus the maximum step length $\eta$, based on a near-on-policy cost in a recent replay buffer $\mathcal{B}_{\text{recent}}$:

$$\arg \min_{0 < \delta \le \bar{\delta}} \delta \times (d - \mathbb{E}_{c_i \in \mathcal{B}_{\text{recent}}} c_i). \tag{19}$$

As a result, the exploration cost is governed by $d$. The adaptive step length tends to fully utilize the budget in safe regions, while remaining conservative in unsafe regions. By using the two lemmas above, we can get the adjusted exploration direction $g^*$ and the step length $\eta^*$. Inserting them back into equation 7 gives the final COX exploration policy.

It should be noted that this section assumes value estimation is accurate, particularly for costs. If the critics cannot provide reliable cost estimates due to the lack of data or function approximation errors, especially in the early stage of training, the data-collection cost is not effectively controlled. Plausible improvements include iusing classical methods, such as reachability analysis (Ganai et al., 2023), or combining COX with model-based RL, such as SafetyDreamer (Huang et al., 2023).

So far, we have explained the "COX-" part, including the effective exploration direction and the adaptive step length under cost constraints. Next, we introduce the "-Q" part about distributional value learning and uncertainty quantification.

## 5 DISTRIBUTIONAL VALUE LEARNING AND UNCERTAINTY QUANTIFICATION

Due to the sparsity of cost and sometimes sparse goal-reaching reward, learning the tails of return and cost distributions is crucial. Therefore, quantile critics are often used in safe RL, for example, in distributional WCSAC (Yang et al., 2023) or ORAC (McCarthy et al., 2025). The objective function in equation 4 indicates that the Bellman update favours overestimation bias of return and underestimation bias of cost (Wu et al., 2024). Moreover, stabilizing the value learning and reducing its gradient variance is important for constraint satisfaction. Considering all these requirements, we adopt Truncated Quantile Critics (TQC) (Kuznetsov et al., 2020).

TQC follows Quantile Regression RL (Dabney et al., 2018). Each independent critic learns the return distribution by a certain number of evenly distributed quantiles. TQC mixes and sorts quantiles from all critics, and then truncates the top $k$ atoms to mitigate the overestimation bias. Specific to safe RL, we truncate the *top $k_r$* atoms for reward and the *bottom $k_c$* atoms for cost critics. The mixed

atoms provide low-variance gradients to stabilize the learning, and the number of truncated atoms controls biases with high flexibility.

Another advantage of TQC is that quantifying distribution-level epistemic uncertainty is straightforward. Suppose that we have $N$ cost critics and $N$ reward critics, each critic predicts $M$ quantiles, denoted as $q_{m,c/r}^{(n)}(s,a)$, representing the quantile function value at level $\tau_m = (m-0.5)/M$. The overall confidence bounds are estimated by computing per-quantile bounds first, and then aggregating them using Conditional Value at Risk (CVaR) (Rockafellar et al., 2000):

$$\hat{q}_{m,c}^l(s,a) = \hat{\mu}_{m,c}(s,a) - \beta_c \hat{\sigma}_{m,c}(s,a), \quad \hat{Q}_c^{\text{LB}}(s,a) = \frac{1}{M} \sum_{m=M-\alpha}^{M} \hat{q}_{m,c}^l(s,a). \tag{20}$$

$$\hat{q}_{m,r}^u(s,a) = \hat{\mu}_{m,r}(s,a) + \beta_r \hat{\sigma}_{m,r}(s,a), \quad \hat{Q}_r^{\text{UB}}(s,a) = \frac{1}{M} \sum_{m=1}^{M} \hat{q}_{m,r}^u(s,a). \tag{21}$$

Here $\hat{\mu}_{m,r/c}$ and $\hat{\sigma}_{m,r/c}$ are quantile-wise mean and standard deviation across $N$ critics, respectively. The two hyperparameters, $\beta_r$ and $\beta_c$, adjust the aggressiveness of exploration. For cost, we use the $\alpha$ head quantiles only, which determines CVaR. A smaller $\alpha$ means more risk-aversion in policy learning, like in WCSAC Yang et al. (2021). For return, we consider the full distribution to compute the optimistic upper bound. Inserting equation 20 and equation 21 into equation 8 gives the corresponding gradients.

Combining COX and TQC-based conservative learning yields the full COX-Q algorithm. It addresses both unconstrained exploration and stable cost value learning in one integrated framework. The implementation is based on SAC (Haarnoja et al., 2018), and keeps the ALM (essentially an enhanced constraint violation penalty) used in CAL (Wu et al., 2024) and ORAC (McCarthy et al., 2025). The pseudo-code of COX-Q is provided in Appendix B.

## 6 EXPERIMENTS

We compare COX-Q against off-policy and on-policy baselines on three safe RL benchmarks:

**Safe Velocity** Safe Velocity is a set of velocity-constrained dense-reward robot locomotion tasks. The agent moves alone in the environment. It has immediate binary cost signals: exceeding the velocity threshold incurs a cost of 1, otherwise 0. The episode cost limit(for 1000 steps) is 5. We select four robots, *hopper*, *walker2d*, *ant*, and *humanoid*, which share the same reward structure. For faster training, experiments are run in Brax (Freeman et al., 2021). Detailed environment settings are explained in Appendix C.1.

**Safe Navigation** Safe Navigation requires a robot to reach a goal or complete a specific task, while avoiding static and moving hazards. The observation is Lidar points. Safe navigation has a small, dense progress reward and a large, sparse goal-reaching reward, along with sparsely activated costs. We select 5 tasks: *SafetyPointGoal2*, *SafetyPointButton2*, *SafetyCarButton1*, *SafetyCarButton2*, and *SafetyPointPush1*, where the suffix "2" denotes the highest difficulty level. In general, controlling a car agent is more difficult than a point agent because a car cannot rotate but steer to the desired location, and button tasks are more challenging than goal tasks. We set $d = 10$. Detailed descriptions about the environment and costs are provided in Appendix C.2.

**SMARTS safe autonomous driving** In autonomous driving, the vehicle interacts with other road users in a closed-loop manner, making it substantially challenging. We use the SMARTS simulation platform (Zhou et al., 2020). Three scenarios with intensive vehicle interactions are selected: *Overtaking* on a two-lane highway, *Intersection* without traffic lights, and *T-junction* without traffic lights. In the last two scenarios, the vehicle needs to execute an unprotected left turn and a lane change sequentially. The reward includes a small distance progress towards the goal and a big bonus if the vehicle reaches the goal. The cost is 10 if the vehicle collides, drives off-road, or violates traffic rules severely (drives into the opposite direction). If the vehicle fails to reach the goal in $60\,\text{s}$, the episode terminates (marked as a timeout). More details are provided in Appendix C.3, including a discussion about the reward and cost design that might be useful for some interested readers.

**Baselines** Selected baselines include representative on-policy and recent state-of-the-art off-policy methods. For on-policy baselines, we select one from each of the categories introduced in Section 2. They are *CUP* (Yang et al., 2022), *RCPO* (Tessler et al., 2018), *PPOSimmerPID* (Sootla et al., 2022b), and *CPPOPID* (Stooke et al., 2020). For Safe Navigation tasks, we replace CPPOPID with TRPOPID because its performance advantage, as shown in Omnisafe (Ji et al., 2024).

Off-policy baselines are more relevant to our proposed method. We choose: (1) *SACLag-UCB* implements conservative double-Q cost learning Hasselt (2010) to augment the SACLag (Ji et al., 2024), which uses SAC with the Lagrangian-based method (Stooke et al., 2020) (2) *CAL* (Wu et al., 2024), which uses the conservative cost learning and ALM (Luenberger et al., 1984). The update-to-data (UTD) ratio is set as 1 to avoid being over-conservative in our sparse-cost tasks; (3) Distributional *WCSAC* Yang et al. (2023), which uses quantile cost critics and a conservative, risk-averse actor objective based on CVaR; (4) *ORAC* McCarthy et al. (2025), a recent method which combines WC-SAC and the ALM in CAL, and further adds risk-averse optimistic exploration towards the low-cost side. Details about the implementations of baselines are explained in Appendix D.

For Safe Velocity and Safe Navigation, we conduct 10 runs with the same 10 random seeds for all methods. For the autonomous driving benchmark, we only select *CPPOPID*, *SACLag*, *CAL*, and *ORAC* as baselines, and only train the policy once with the same random seed due to the long training time. The code is available via: https://github.com/RomainLITUD/COXQ

## 6.1 RESULTS ON SAFE VELOCITY

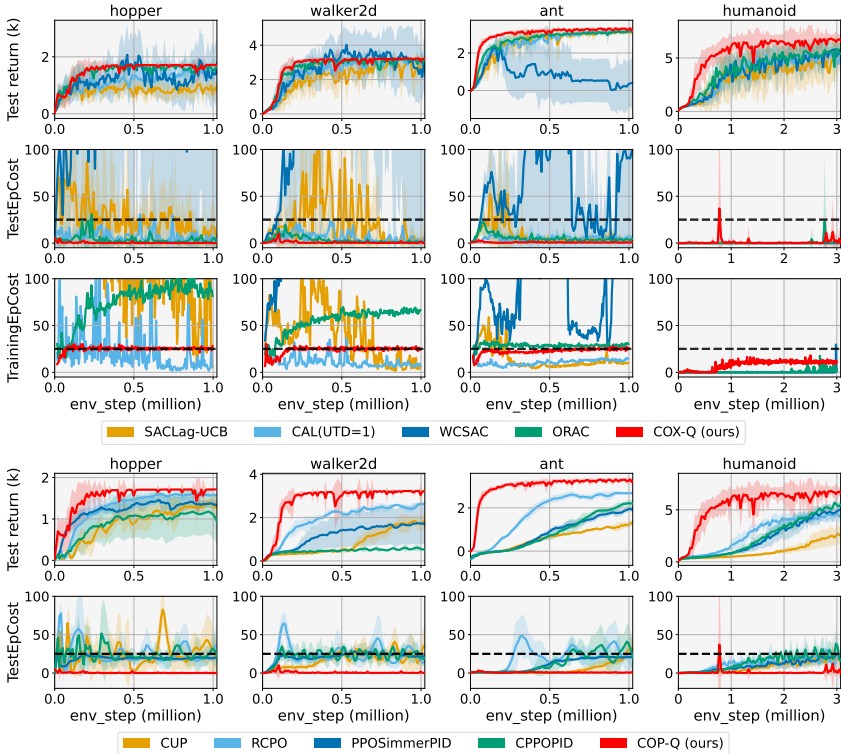

Figure 1: COX-Q v.s. off-policy (top) and on-policy (bottom) baselines. TrainingEpCost is for data collection, which is expected to stay near or below the threshold throughout the training. Note that training and test costs are identical for on-policy baselines.

The results on the Safe Velocity benchmark are presented in Figure 1. Overall, COX-Q demonstrates superior sample efficiency, achieves high cumulative returns, and has nearly-zero test costs fast, while keeping data collection costs below the predefined budget. More specifically: (1) COX-Q exhibits a clear advantage in data efficiency over on-policy baselines. Its off-policy nature and the truncation mechanism in TQC enable a test cost significantly lower than the budget, a property

that on-policy methods do not have. (2) By comparing COX-Q against *SACUCB-PID* and *CAL*, we observe that distributional RL has higher sample efficiency than point-value based baselines. (3) The cost-constrained exploration and step-length auto-tuning effectively regulate the data-collection cost, especially in the middle and late training phases. This is evidenced by the smooth and horizontal (near the threshold) training cost profiles of COX-Q in all tasks. In contrast, baseline methods incur higher training costs on one or more tasks due to unregulated optimistic exploration. For *humanoid*, no baseline policies can make the robot walk fast enough to reach the unsafe boundary, so both training and test costs are near-zero. These observations highlight the key strengths of COX-Q: high data efficiency, improved safety, and controlled training costs for exploration. Next, we assess its performance in exploration-challenging environments.

## 6.2 RESULTS ON SAFE NAVIGATION

Due to the sparse goal-reaching rewards and costs in Safe Navigation, truncating too many atoms can suppress the learning progress. Therefore, we preserve the mixed quantiles in COX-Q but do not apply truncation. Instead, we use the estimated CVaR-based upper bound of cost to update the actor and Lagrangian multiplier, same as in Worst-Case SAC (Yang et al., 2021).

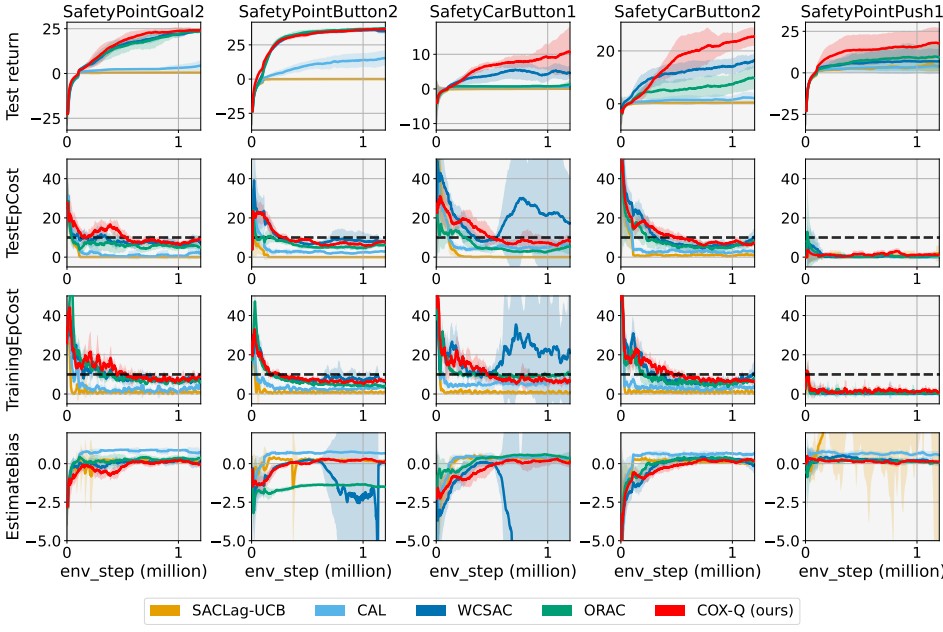

Figure 2: Benchmark of COX-Q against off-policy baselines on safe navigation tasks (episode cost limit is 10). The bottom figure is the cost value estimation bias, computed from cost critic outputs and the recorded trajectories in the evaluation phase. Below 0 means underestimation.

The results are summarized in Figure 2. For on-policy baselines, the results are provided in the Appendix E. In general, COX-Q achieves on-par or higher returns than baselines. The advantage is significant in more challenging tasks *CarButton1*, *CarButton2*, and *PointPush1*. The training and test costs both converge below the limit. The estimation bias of COX-Q consistently converges to 0 with training, while all baselines are either over-conservative or unstable. This observation indicates that the mixed quantiles significantly improve value learning.

## 6.3 ABLATION STUDIES ON SAFE VELOCITY AND SAFE NAVIGATION

Next, we evaluate the contribution and role of each module of COX-Q. Two variants are compared: (1) with TQC only, without exploration; (2) TQC + ORAC-style exploration.

The results on 4 selected tasks are shown in Figure 3. In all tasks, all ablation models' returns remain higher than baselines, which indicates that the TQC contributes mainly to improved returns. In Safe

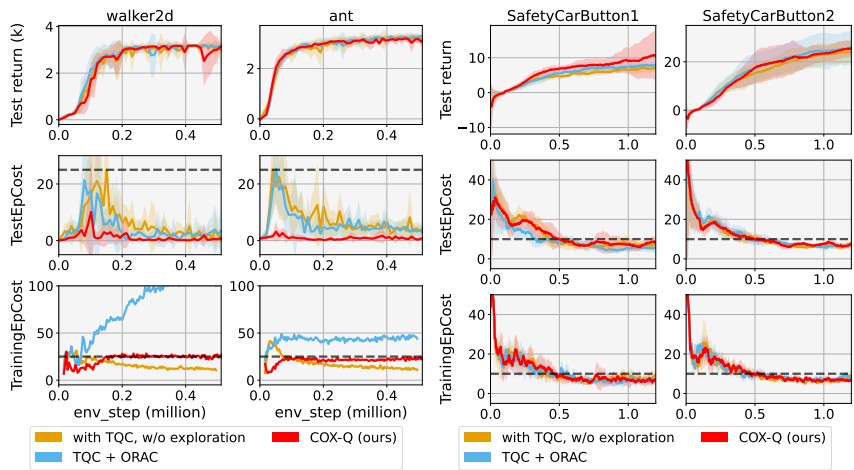

Figure 3: Ablations on Safe Velocity and Safe Navigation.

Velocity, the exploration strategy significantly influences training costs. Both ORAC and COX-Q exploration increase training cost, but the proposed cost-constrained mechanism well controls training costs below the given budget. However, in Safe Navigation, COX-Q and ORAC exploration exhibit close performances to the TQC-only baseline. This highlights two properties of the task: (1) Gradient conflict in the action space is weak in Safe Navigation due to the sparse obstacles. Therefore, ORAC and COX-Q become identical in most cases. In Appendix E, the analysis shows that the ratio of triggered gradient conflicts in the first 200K steps is below 10%, and even below 2% for PointPush1. (2) Cost value learning in Safe Navigation is highly biased due to the sparsity of cost signals. As shown by the bottom row of Figure 2, the cost is underestimated during the early training stage. Correspondingly, the cost constraint violations for training and testing are triggered. This important observation indicates that, for constrained RL with sparse costs, the underestimation bias in the cumulative cost is the major bottleneck, rather than the exploration mechanism. Possible improvements include inducing Hindsight Experience Replay (HER) (Andrychowicz et al., 2017) or prioritized experience replay (Schaul et al., 2015) to better estimate the cost objective.

## 6.4 EVALUATION ON SMARTS SAFE AUTONOMOUS DRIVING

Finally, we evaluate the effectiveness of COX-Q in more challenging autonomous driving tasks, in which surrounding vehicles have closed-loop interactions with the controlled RL agent. Autonomous driving is a typical safety-critical task. We set a nearly-zero cost limit (0.01), same as in SafetyDreamer's MetaDrive task (Huang et al., 2023). The vehicle stays in "unsafe" regions (the cumulative cost is above 0.01) during data collection and aims to minimize the test cost as much as possible, thus intentionally increasing the frequency of exploration gradient conflict and the proportion of constrained exploration. Because the policy stays in unsafe regions during training, we do not add the step length auto-tuning in equation 19 to avoid the exploration converging to zero, which will make COX-Q the same as original TQC. Additionally, in this benchmark, we use the TQC-based ORAC to focus on the differences caused by the exploration mechanism. After 512K steps of training, we run 2000 episodes with stochastic initial states to obtain the test performance.

The test performance is presented in Table 1, and the number of unsafe events (collisions and off-road) during training is listed in Table 2. Overall, COX-Q achieves the best safety performance in tests without incurring significant excessive exploration cost or exhibiting over-conservative driving behaviours (time-out). Moreover, compared to ORAC, COX-Q significantly reduces both unsafe events during data collection and timeouts during testing. This observation indicates that resolving conflicting gradients in a direction that simultaneously reduces cost and improves reward can effectively maximize return while avoiding being over-conservative. Another notable point is that the safety performance of all methods in *overtaking* is relatively worse. The reason is that SMARTS uses an instantaneous lane change model from SUMO (Krajzewicz et al., 2012), making collision avoidance inherently hard due to the lack of warning (e.g., turn signals).

Table 1: Test safety performance on SMARTS (512K steps, 2000 stochastic runs)

| Scenario | Metric | CPPOPID | SACLag | CAL | TQC-ORAC | COX-Q (ours) |
|---|---|---|---|---|---|---|
| Overtaking | Collision | 331 | 194 | 186 | 97 | 99 |
| | Off-road | 96 | 2 | 7 | 3 | 4 |
| | Rule violation | 3 | 0 | 0 | 0 | 0 |
| | Timeout | 0 | 2 | 1 | 887 | 0 |
| Intersection | Collision | 183 | 33 | 23 | 18 | 12 |
| | Off-road | 22 | 2 | 1 | 1 | 2 |
| | Rule violation | 9 | 18 | 0 | 0 | 0 |
| | Timeout | 0 | 0 | 1 | 12 | 0 |
| T-junction | Collision | 195 | 55 | 36 | 28 | 21 |
| | Off-road | 91 | 2 | 0 | 5 | 0 |
| | Rule violation | 3 | 24 | 0 | 0 | 0 |
| | Timeout | 0 | 0 | 17 | 86 | 5 |

Table 2: Number of unsafe events in data collection (512K steps, excluding the initial 5120 steps)

| Scenario | CPPOPID | SACLag | CAL | TQC-ORAC | COX-Q (ours) |
|---|---|---|---|---|---|
| Overtaking | 3697 | 1570 | 1544 | 3215 | 1665 |
| Intersection | 4969 | 1755 | 739 | 3589 | 1123 |
| T-junction | 5513 | 1965 | 1675 | 3837 | 1794 |

## 7 CONCLUSIONS

This paper proposes an off-policy primal-dual safe RL method, constrained optimistic exploration Q-learning, involving a cost-constrained optimistic exploration strategy and TQC-based conservative value learning. The proposed COX-Q is evaluated in three representative safe RL benchmarks. The results demonstrate that COX-Q has significantly higher data efficiency than on-policy baselines. When the exploration gradient conflict between reward and cost is significant (Safe Velocity and SMARTS), COX-Q shows superior safe performance in tests, while effectively controlling exploration cost in data collection. When the exploration gradient conflict is weak, or the bias in cost estimation is high due to sparse cost signals (Safe Navigation), COX-Q is on par or better with the state-of-the-art method. In addition, the autonomous driving experiment showcases that the proposed method can be used in complex environments with large neural networks. In conclusion, COX-Q is a promising solution to RL applications with data efficiency and safety concerns.

**Limitations** The major limitation of this study is the reliability of quantified epistemic uncertainty. TQC mixes quantiles from all critics and learns the entire return distribution. Therefore, the diversity of critics for nearly Out-Of-Distribution samples might be suppressed due to highly correlated gradients for all critics. Implementing improved methods such as diverse ensemble projection (Zanger et al., 2023) or random priors (Osband et al., 2018) to enhance the quality of epistemic uncertainty quantification is a potential future research direction. Another future research direction is how to effectively implement COX in sparse-cost tasks such as SafeNavigation. A key step is to use, e.g., HER (Andrychowicz et al., 2017) or prioritized experience replay (Schaul et al., 2015) to robustify the cost-critic learning.

**Acknowledgments** This work has received funding from the European Union's Horizon 2020 Research and Innovation program under Grant Agreement No. 964505 (E-pi).

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

# A  Proofs of the two lemmas

## A.1  Lemma-1

The solution of equation 11 and equation 12 is given as follows. For simplicity, we denote $g_1 = g_r$, $g_2 = -g_c$, and $\Sigma = \Sigma_T$ here.

Let $S = \text{span}\{g_1, g_2\}$. Decompose $g_t = g_S + g_\perp$ with $g_S \in S$ and $\langle g_\perp, g_1 \rangle_\Sigma = \langle g_\perp, g_2 \rangle_\Sigma = 0$. Constraints depend only on $g_S$. Therefore, it suffices to solve in $S$ and then add back $g_\perp$. This becomes a 2D problem. We next derive the KKT conditions. With the inequalities $c_1(u) = -\langle g_1, u \rangle_\Sigma \leq 0$ and $c_2(u) = -\langle g_2, u \rangle_\Sigma \leq 0$, we add two *non-negative* multipliers to form the Lagrangian:

$$\mathcal{L}(u, \mu_1, \mu_2) = \frac{1}{2}\|u - g_t\|_\Sigma^2 + \mu_1(-\langle g_1, u \rangle_\Sigma) + \mu_2(-\langle g_2, u \rangle_\Sigma). \tag{A.1}$$

Then the stationarity is:

$$\nabla_u \mathcal{L} = \Sigma(u - g_t) - \mu_1 \Sigma g_1 - \mu_2 \Sigma g_2 = 0 \quad \Rightarrow \quad u = g_t + \mu_1 g_1 + \mu_2 g_2 \tag{A.2}$$

The primal feasibility gives:

$$\langle g_1, u \rangle_\Sigma \geq 0, \quad \langle g_2, u \rangle_\Sigma \geq 0. \tag{A.3}$$

The complementary slackness gives

$$\mu_1 \langle g_1, u \rangle_\Sigma = 0, \quad \mu_2 \langle g_2, u \rangle_\Sigma = 0. \tag{A.4}$$

So, we define the so-called $\Sigma$-Gram scalars and target correlations as:

$$s_{ij} = \langle g_i, g_j \rangle_\Sigma, \quad v_i = \langle g_i, g_t \rangle_\Sigma, \tag{A.5}$$

and plug stationarity into the constraints:

$$\begin{bmatrix} s_{11} & s_{12} \\ s_{21} & s_{22} \end{bmatrix} \begin{bmatrix} \mu_1 \\ \mu_2 \end{bmatrix} = -\begin{bmatrix} v_1 \\ v_2 \end{bmatrix} \tag{A.6}$$

Because the Gram matrix is apparently SPD if $g_1$ and $g_2$ are not co-linear, the solution is unique whenever both constraints are active. For the degenerated co-linear cases, we assume that $g_1 = \alpha g_2$. If $\alpha > 0$, then $K$ is a half-space, then the solution is a direct projection:

$$g^* = u = g_t - \min(0, \frac{v_1}{s_{11}})g_1. \tag{A.7}$$

If $\alpha < 0$, constraints reduce to $\langle g_1, u \rangle_\Sigma = 0$ (hyper-plane):

$$g^* = u = g_t - \frac{v_1}{s_{11}}g_1, \tag{A.8}$$

and $\alpha = 0$ is trivial.

For non-degenerate cases, we apply the optimal active set $A = \{c_1(u), c_2(u)\}$. There are four possibilities:

(1) No constraint active: Then $\mu_1 = \mu_2 = 0$, $g^* = g_t$.

(2) Only $c_1(u)$ active: Set $\mu_2 = 0$. From $\langle g_1, u \rangle_\Sigma = 0$, we get:

$$\mu_1 = -\frac{v_1}{s_{11}} \quad \Rightarrow \quad u^* = g_t - \frac{v_1}{s_{11}}g_1. \tag{A.9}$$

(3) Only $c_2(u)$ active: Similar to the previous case, we have:

$$u^* = g_t - \frac{v_2}{s_{22}}g_2. \tag{A.10}$$

(4) Both boundaries active: Then we solve equation A.6. That gives:

$$\mu_1 = \frac{-s_{22}v_1 + s_{12}v_2}{\det G}, \quad \mu_2 = \frac{s_{12}v_1 - s_{11}v_2}{\det G}, \quad \det G = s_{11}s_{22} - s_{12}^2 \geq 0 \tag{A.11}$$

$$u^* = g_t - \mu_1 g_1 - \mu_2 g_2. \tag{A.12}$$

Replace $g_1$ and $g_2$ by $g_r$ and $-g_c$, respectively, then the proof of Lemma 1 is done.

## A.2 LEMMA-2

The solution of equation 16 is derived based on two cases;

*Case A:* $s < 0$, which means moving along the $u^*$ does not increase the expected cost. The hinge $\phi(\eta)$ is non-increasing w.r.t. $\eta$. Therefore, minimal violation is achieved by taking the largest trust region:

$$\eta^* = \eta_{\text{KL}} \tag{A.13}$$

*Case B:* $s > 0$, which means moving along the $u^*$ increase the expected cost. Then we check the feasibility of a zero-violation set on the ray $\{\eta : \eta s \leq r\}$. If $r \leq 0$, then the zero-violation set is empty on $[0, \eta_{\text{KL}}]$ and the hinge increases with $\eta$. Therefore, the minimizer is trivial $\eta^* = 0$. If $r \geq 0$, simply take the boundary as the zero-violation set:

$$\eta^* = \min(\eta_{\text{KL}}, \frac{r}{s}) \tag{A.14}$$

*Case C:* $s = 0$, which means the hinge becomes a constant. In this case. If $r \geq 0$, every $\eta \in [0, \eta_{\text{KL}}]$ is optimal. If $r < 0$, violation is unavoidable. We set $\eta^* = 0$ by rule for conservativeness.

Combining the three cases above gives the complete proof of Lemma 2.

# B  IMPLEMENTATION DETAILS OF COX-Q

---

**Algorithm 1** COX-Q based on SAC, with optional Augmented Lagrangian Method (ALM)

---

**Input and initialization:** policy network $\pi_\theta(s)$, $N$ reward quantile critic networks $\{q_{\psi_i,r}\}_{i=1}^N$, $N$ cost quantile critic networks $\{q_{\psi_i,c}\}_{i=1}^N$, with $M$ quantile heads.
replay buffer $\mathcal{D}$, truncation parameters $k_r$ and $k_c$, exploration optimism parameters $\beta_r$ and $\beta_c$, cost limit $d$, maximum trust region size $\eta_{\text{KL}}$ ($\delta$), Lagrangian multiplier $\lambda$,
risk-level CVaR $\alpha$
**repeat**
    Observe State $s_t$,
    **if** use COX **then**
        Compute the target policy $\mathcal{N}(\mu_T, \Sigma_T) = \pi(s_t)$
        Compute $\hat{Q}_r^{\text{UB}}$, $\hat{Q}_c^{\text{LB}}$, $\hat{Q}_c^{\text{mean}}$ from critics using equation 20 and equation 21
        Compute their gradients $g_r, g_c, g_m$ w.r.t $\mu_T$
        **if** $\hat{Q}_c^{\text{mean}}$ in safe area **then**
            compute $g^* = g_t = g_r - \lambda g_c$
        **else**
            Compute aligned exploration gradient $g^*$ using equation 14
        **end if**
        Compute adjusted step length $\eta^*$ using equation 18.
        Compute action shift $\mu_\Delta$ using OAC formula from $\eta^*$ and $g^*$
        select action $a_t = \text{clip}(\mu_e + \epsilon, a_{\text{lower}}, a_{\text{upper}})$, where $\epsilon \sim \mathcal{N}(\mu_\Delta, \Sigma_t)$
    **else**
        select action $a_t = \text{clip}(\mu_\theta(s_t) + \epsilon, a_{\text{lower}}, a_{\text{upper}})$, where $\epsilon \sim \mathcal{N}(0, \Sigma_t)$
    **end if**
    Execute $a_t$, observe next state $s_{t+1}$, reward $r_t$ and cost $c_t$
    Store the transition $(s_t, a_t, (r_t, c_t), s_{t+1})$ in $\mathcal{D}$
    **if** critic/actor update **then**
        Execute TQC or Worst-Case SAC updates, with optional ALM (used by default)
    **end if**
    **if** $\eta_{\text{KL}}$ update **then**
        Sample a recent $N_r$ transitions from $\mathcal{D}$, compute the average cost
        Update $\delta$ using equation 19.
    **end if**
**until** Convergence

---

In the pseudo-code of Algorithm 1, the updates of critics are the same as the original TQC (Kuznetsov et al., 2020). The actor update involves the ALM proposed by Luenberger et al. (1984) and introduced in safe RL by Wu et al. (2024). ALM alters the optimization objective of the actor by the following equations:

$$
\begin{cases}
\max_\pi \mathbb{E}_{s \sim \rho_\pi, a \sim \pi(\cdot|s)}[\hat{Q}_r^{\text{mean}} - \lambda(\hat{Q}_c^{\text{UB}} - d) - \frac{c}{2}(\hat{Q}_c^{\text{UB}} - d)^2], & \text{if } \ \frac{\lambda}{c} \geq d - \mathbb{E}(\hat{\mathbb{Q}}^{\text{UCB}}) \\
\max_\pi \mathbb{E}_{s \sim \rho_\pi, a \sim \pi(\cdot|s)}(\hat{Q}_r^{\text{mean}}), & \text{otherwise}
\end{cases}
\tag{B.1}
$$

The added quadratic term helps conform to cost constraints and move the optimization direction towards the cost limit, which can accelerate the learning process. In our studies, we use $c = 10$ for all tasks. This ALM is used for CAL, ORAC, and COX-Q in all experiments, as in their original paper.

In addition, off-policy safe RL needs to set the cap on Q-values $d$ in an "on-policy" approach, instead of directly using the test episode costs as in on-policy methods. This is explained in the paper of CVPO (Liu et al., 2022), using the following formula:

$$
d = d_{episode} \frac{1 - \gamma^T}{T(1 - \gamma)},
\tag{B.2}
$$

in which $T$ is the episode length. In all off-policy methods used in this study, we use this formula to convert the episode cost limit to the limit on $Q_c^\pi$.

## C  DESCRIPTION OF THE THREE SAFE RL ENVIRONMENTS

### C.1  SAFETYVELOCITY-V1



Figure C.1: The four selected robots in SafetyVelocity-v1 benchmark.

For the selected 4 robots, their configurations are shown in Figure C.1. They share the same reward structure as follows:

$$
r_t = w_h \times r_{\text{health}} + w_v \times r_{\text{velocity}} - w_c \times r_{\text{ctrl}},
\tag{C.1}
$$

in which $r_{\text{health}}$ is a binary reward. If the robot keeps upright, get +1 reward; otherwise, get 0 and terminate the episode. $r_{\text{velocity}}$ is a reward equal to the moving velocity along a given direction. $r_{\text{ctrl}}$ is the control cost penalty, measuring how much torques are applied to the joints. $w_h$, $w_v$ and $w_c$ are three positive weights. Cost is binary. For hopper and walker2d, if the velocity along the +x axis exceeds the threshold, the cost is +1; otherwise, 0. For ant and humanoid, if the velocity along any direction exceeds the threshold, the cost is +1; otherwise, 0. The episodic cost limit is set to 25, as recommended in the original paper (Zhang et al., 2020). The weight coefficients, velocity thresholds, and the dimensionality of action spaces for different robots are listed in Table C.1. All implementations are based on the Brax (Freeman et al., 2021), using the same parameters (e.g. velocity thresholds) as in Safety-Gymnasium (Ji et al., 2023). Brax supports fully parallelized simulations on GPU, so it can save a lot of time for training. The default "generalized" backend is used for simulation.

### C.2  SAFE NAVIGATION IN SAFETY-GYMNASIUM

In Safe Navigation, the name of a task is composed of two parts. "-Point-" or "-Car-" in the middle indicates what is the type of robot used, as shown on the top of Figure C.2. *Point* is a simple robot that has two actuators, one for rotation and the other for forward/backward movement. *Car* is a

Table C.1: Weight coefficients and velocity threshold for SafetyVelocity-v1

| ROBOT | $(w_h, w_v, w_c)$ | velocity threshold | Action dimension |
|---|---|---|---|
| hopper | (1, 1, 0.001) | 0.7402 | 3 |
| walker2d | (1, 1, 0.001) | 2.3415 | 6 |
| ant | (1, 1, 0.5) | 2.6222 | 8 |
| humanoid | (5, 1.25, 0.1) | 1.4119 | 17 |

more complex robot that can move in two dimensions. It is equipped with two independently driven parallel wheels and a freely rotating rear wheel. Both steering and forward/backward motion require coordinated control of the two drive wheels, imposing more complex control dynamics. Both robots are equipped with 2D Lidars to perceive the environment. Their action dimensionalities are both 2.

The last part of the name indicates the type of task and its difficulty level. Three example tasks are shown at the bottom of Figure C.2.

- *Goal2:* The robot needs to reach a goal position (green pillar) while avoiding touching hazard pitfalls (blue circles) or move fragile vases (while cubes).

- *Button2:* The robot needs to reach the correct button (orange spheres) among 4 buttons, while avoiding touching blue-circle pitfalls or being hit by the moving gremlins (purple cubes moving in a circle).

- *Push1:* The robot needs to push the yellow object to the green goal position while avoiding blue pitfalls and the tall pillar.

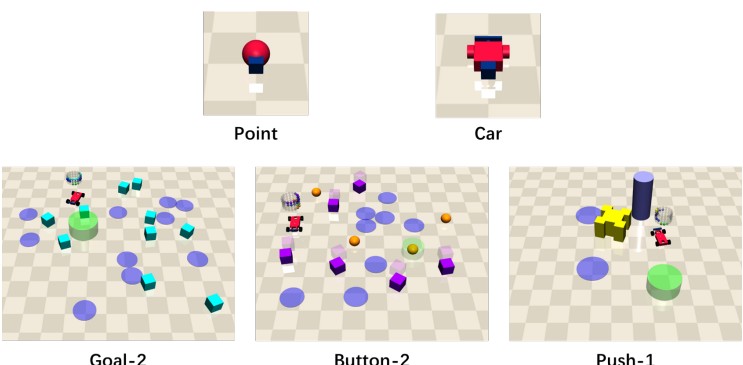

Figure C.2: The robots and the tasks in the safe navigation benchmark.

The reward and cost designs are complicated, depending on each specific task. We refer the readers to the public webpage of the Safety-Gymnasium for more details: https://safety-gymnasium.readthedocs.io/en/latest/environments/safe_navigation.html.

Additionally, to accelerate the learning process, the simulation time step is modified to 2.5 times the original value, according to the paper of CVPO (Liu et al., 2022) and CAL (Wu et al., 2024). While ORAC (McCarthy et al., 2025) does not release its code, the final reward performance implies that they probably used the same simulation settings. We therefore also keep the modification.

## C.3 SMARTS AUTONOMOUS DRIVING

SMARTS is a scalable RL training platform for autonomous driving (Zhou et al., 2020), providing closed-loop simulation in diverse traffic scenarios. In this paper, we control an ego vehicle (red) to drive through the scenario. The ego vehicle has two actions: accelerations (between $\pm 6.5\,\mathrm{m\,s^{-2}}$) and steering rate (between $\pm 1.5\,\mathrm{rad\,s^{-1}}$ for intersections and $\pm 0.7\,\mathrm{rad\,s^{-1}}$ for highways). Then the vehicle's motion is controlled by a bicycle model (Gillespie, 2021). In the simulation, the vehicle

can only change its actions every $0.25\,\mathrm{s}$ to avoid oscillating trajectories. Note that our settings are more realistic than the original SMARTS. In their default action spaces, the ego vehicle has infinite acceleration and can completely stop from the highest speed in $0.1\,\mathrm{s}$.

The three scenarios are illustrated in Figure C.3. For the intersection and the T-junction, the ego vehicle needs to first pass an unsignalized area and execute an unprotected left turn, then change to the right lane to reach the goal. For highway over-taking, the leading vehicle is slow, and other vehicles can change their lanes arbitrarily. The ego vehicle needs to overtake the slow vehicle and reach the goal on the same lane. All surrounding traffic vehicles are controlled by a set of predefined driving models with a distribution of inner parameters, providing diverse interactions.

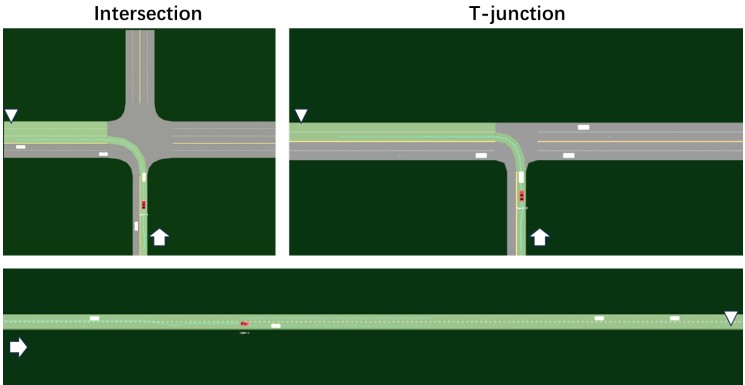

Figure C.3: The three autonomous driving scenarios in SMARTS benchmarks. Arrows are the entering lane of the ego vehicle, and triangles are goal positions. Highlighted green lanes are the "on-route" areas for the ego vehicle. White boxes are surrounding traffic vehicles.

The reward and cost design follows the minimalist principle:

$$R = r_{\mathrm{distance}} + r_{\mathrm{goal}}. \tag{C.2}$$

The first term is the travelled distance (in meters) within one decision step ($0.25\,\mathrm{s}$). The second term is +30 if reaching the goal. The cost is 0 when staying safe. When collisions, off-road, driving on the wrong side of the road, or off-route happen, the cost is +10. The first three situations also trigger the termination of the episode.

We hereby give a short discussion about our reward and cost design that might be useful for interested readers. We actually tried many other different designs, but this simplest version works the best. The observed issues of other settings are summarized below:

- *Do not terminate the episode when an unsafe event happens:* This is similar to the method used in Safety Dreamer's MetaDrive task (Huang et al., 2023). However, in our intersection and T-junction scenarios, due to the complexity of the road layout, the replay buffer is filled with meaningless, unsafe cases in the early stage of training. For example, when the ego vehicle drives off-road, it may stay there for a long time until the episode ends. This severely hinders policy learning.

- *Assign different costs to different unsafe events:* Many RL studies on autonomous driving tasks, e.g., MetaDrive (Li et al., 2022), give a higher penalty for severe events like collisions, and a smaller penalty for traffic rule violations. In our trials, we found that the agent tends to do "reward-hacking" in such settings. For example, the vehicle will choose to drive off-road to get a lower penalty instead of learning how to avoid collisions. This reward-hacking is particularly severe when the vehicle needs to do a series of actions to solve the final potential collision, as is our case (restricting the acceleration and steering rate).

- *Use risk field or Surrogate Safety Measures (SSMs) as costs:* Using SSMs (Wang et al., 2021), such as Time-to-Collision (TTC) or risk field, to shape the reward is also a widely-used technique in RL-based autonomous driving. Our trials found that using TTC and

the capsule risk field can indeed accelerate learning in the early stage. However, the final performance is worse than our simplest setting. One of the possible reasons could be that these SSMs add inductive biases to safety. They focus on one or several specific types of unsafe (potential collision) cases. This may restrict the exploration power of RL. The simple end-oriented costs, in contrast, can encourage exploring diverse and better solutions.

Both policy and critic networks by default use the WayFormer (Nayakanti et al., 2023) structure. For reward and cost critics, they share the torso and use different MLP heads to give multiple predictions of returns. Their network structures are briefly illustrated in Figure C.4.

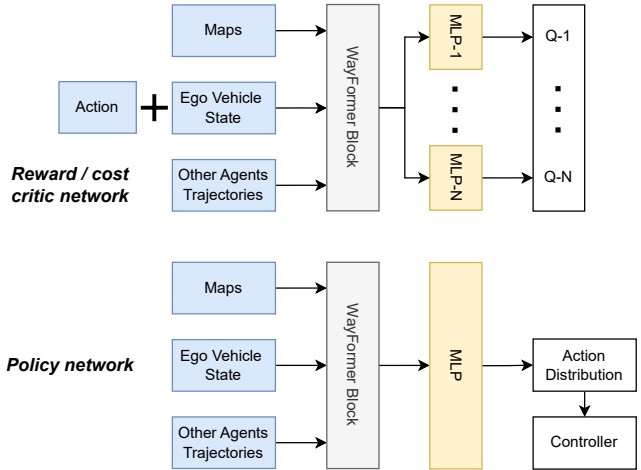

Figure C.4: The policy and critic network structure for SMARTS.

## D    HYPERPARAMETER SETTINGS

For on-policy baselines, we use the same 1M step hyperparameter settings recommended by the OmniSafe benchmark platform (Ji et al., 2024) for all experiments. Details are provided on their public webpage https://github.com/PKU-Alignment/omnisafe. We did some modifications to make the training faster, which are available in the open-source code.

For COX-Q, the implementation is based on SAC (Haarnoja et al., 2018). The shared parameters are listed in Table D.1, and the environment-specific parameters are listed in Table D.2. SACLag-UCB was implemented based on the SACLag method provided in OmniSafe. For CAL (Wu et al., 2024), we use the same hyperparameters in the original paper, except for the randomized ensemble technique and the UTD ratio (1 in our experiments). It is useful to note that the original CAL paper uses UTD=20 in their experiments with densified cost signals (e.g., in the Safe Velocity experiments, the cost is not 0/1 but the real-time speed). While in our 0/1 sparse cost settings, using high UTD will quickly make the policy over-conservative in training, thus suppressing the maximization of return. This explains why we choose UTD=1.

The code of ORAC (McCarthy et al., 2025) is not available yet. For safe navigation tasks, we use the recommended hyperparameters in the ORAC paper in our own implementation. While for Safe Velocity and SMARTS, we did not find a proper set of hyperparameters for the original ORAC. The performance is quite unstable. Therefore, we choose to modify ORAC based on our quantile critics implementation. For all off-policy methods, we use the same discount factor and episode length listed in Table D.2 for consistency.

To accelerate the training for Safe Velocity and SMARTS, we use a high number of parallel environments and a lower offline update frequency, as recommended by Brax (Freeman et al., 2021).

Table D.1: Shared off-policy parameters

| Parameters | Value) |
|---|---|
| Policy learning rate | 3e-4 |
| Critic learning rate | 3e-4 |
| Entropy learning rate | 3e-4 |
| Batch size | 256 |
| Maximum step length $\eta_{\mathrm{KL}}$ | 6 |
| Tau | 0.005 |
| Convexification $c$ in ALM | 10 |
| Number of cost critics | 5 |
| Number of reward critics | 5 |

Table D.2: Environment-specific off-policy parameters

| Parameters | Safe Velocity | Safe Navigation | SMARTS |
|---|---|---|---|
| Episode length | 1000 | 400 | 240 |
| discount factor $\gamma$ | 0.99 | 0.975 | 0.975 |
| Episode cost limit | 25 | 10 | 0.01 |
| Number of parallel envs | 64 | 1 | 128 |
| Gradient steps | 64 | 1 | 64 |
| Policy update steps | 64 | 1 | 64 |
| Lagrangian initial value | 1 | 0.001 | 1 |
| Lagrangian learning rate | 3e-4 | 5e-4 | 3e-4 |
| Step length auto-tuning learning rate | 1e-4 | 1e-4 | NA |
| Initial steps | 10240 | 5000 | 5120 |
| Buffer size | 1024000 | 1000000 | 512000 |
| Policy network | $256 \times 2$ | $256 \times 2$ | Wayformer |
| Critic network | $256 \times 5$ | $256 \times 2$ | Wayformer |
| Layer Normalization | False | False | NA |
| Entropy auto-tuning | True | False | True |
| Number of quantiles $M$ | 25 | 32 | 25 |
| Truncation $(k_r, k_c)$ | (2, 5) | (0, 0) | (1, 0) |
| Optimism $(\beta_r, \beta_c)$ | (4, 3) | (3, 3) | (3, 3) |
| Cost CVaR $\alpha$ | 13 | 16 | 13 |
| Target update frequency | 64 | 2 | 64 |

# E   SUPPLEMENTARY RESULTS

The performance of COX-Q against on-policy baselines on safe navigation tasks is presented in Figure E.1. Although they adhere to the cost constraints, the returns are significantly lower than off-policy baselines.

Figure E.2 gives the percentage of triggered exploration gradient conflicts for the first 200K steps in the safe navigation benchmark using COX-Q. We see that the reward and cost objectives rarely conflict with each other ($< 10\%$); Additionally, as shown in Figure 2, the policy starts from unsafe regions in the early stage of training, which will adjust the exploration step length to a small value. Therefore, the differences between ORAC and COX-Q are small. We hereby give two possible explanations: First, just like in conventional multi-task learning, the gradient conflicts often happen between two loss functions with significantly different scales. However, for safe navigation, both reward and cost are on the same scale (0-30). Second, as both reward and cost are sparse signals (or at least highly skewed), most exploration gradients are near zero, making it highly stochastic.

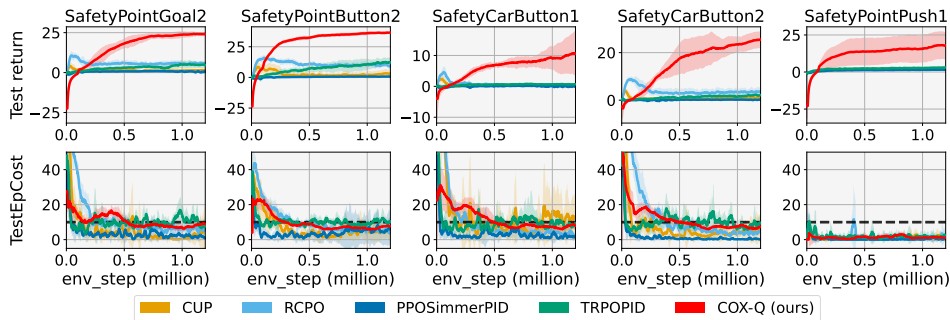

Figure E.1: Training curves of on-policy baselines and COX-Q for safe navigation tasks

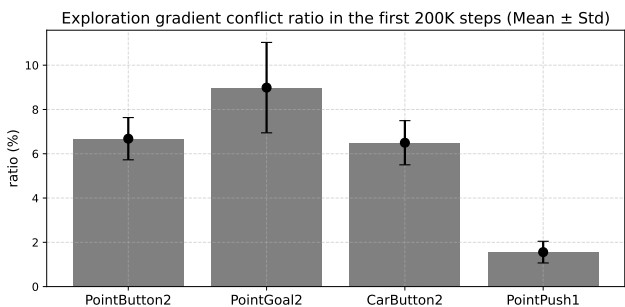

Figure E.2: Exploration gradient conflict analysis for safe navigation tasks

## F  THE USE OF LARGE LANGUAGE MODELS (LLMS)

LLMs are used for polishing writing only, such as selecting proper words.

