# OpenReview forum: "Off-Policy Safe Reinforcement Learning with Cost-Constrained Optimistic Exploration"
_ICLR.cc/2026/Conference — ICLR 2026 Poster_

### Official Review · Reviewer_ARdC · 2025-10-25

**Soundness:** 2
**Presentation:** 1
**Contribution:** 1
**Rating:** 0
**Confidence:** 4

**Summary:**

This paper introduces an off-policy primal–dual safe reinforcement learning (RL) algorithm named COX-Q, aiming to improve sample efficiency and reduce estimation bias. The method integrates a cost-constrained optimistic exploration strategy to adjust the trust region for safe exploration. Additionally, the authors propose to mitigate estimation bias through the use of truncated quantile critics.

**Strengths:**

The empirical evaluations are comprehensive, with three applications.

**Weaknesses:**

1. The presentation of the theoretical results is weak. The paper does not clearly state the assumptions underlying the proposed method, leaving it unclear under what conditions the approach is valid or applicable. The policy’s action distribution is simply assumed to be Gaussian without justification. Moreover, Lemma 1 and Lemma 2 are rather trivial and do not appear to offer substantial theoretical contribution. If these are presented as lemmas, one would expect a main theorem or stronger result to follow—otherwise, the theoretical section lacks depth.
2. Even if the lemmas are accepted, the authors fail to explain how these results contribute to improving sample efficiency or mitigating estimation bias. While the empirical evaluations suggest some performance gains, it remains unclear under what circumstances the proposed method is effective. The framework does not appear to generalize well.
3. Although the experimental coverage is broad, the presentation quality is poor. For instance, no quantitative results are provided to support the claim of reduced estimation bias. In addition, the explanations are poorly organized and often lack logical coherence, which makes the empirical section difficult to follow.
4. There is no pseudo code for clarification and implementation.

**Questions:**

Given the major conceptual and presentation issues identified above, I do not have specific technical questions for the authors. The paper requires substantial clarification and restructuring.

---

> ### Author Response · Authors · 2025-11-22
> **Official Responses of Submission4892 to Reviewer ARdC - Part 1**
>
> We thank the reviewer for taking the time to read our manuscript and provide feedback. We believe the score of 0 (very strong rejection) suggests that there may be substantial misunderstandings or missing context regarding our contributions. We do our best to clarify these points, explain the organization and presentation choices of the paper, and address the specific concerns raised. Our detailed, point-by-point responses are given below.
>
> &nbsp;
>
> **Q1. About the assumption/applicability of the theory. And the Gaussian policy assumption.**
>
> Our theoretical Section 4 focuses specifically on cost-constrained optimistic exploration during data collection. The goal is to guarantee that the data-collection safety cost respects the given threshold.
>
> The only underlying "assumption" here is that the action distribution is diagonal Gaussian. In our setting, this is not a probabilistic modelling assumption about the environment or labels, but rather **a design choice for exploration**. The purpose of the Gaussian is to generate local perturbations around the mean policy action to induce stochastic exploration. **Diagonal Gaussian policy is standard and widely adopted in deep RL.** Off-policy methods such as DDPG, TD3, and SAC typically use Gaussian noise in the action space. On-policy methods such as PPO also use Gaussian policies for continuous actions. Our theoretical development builds on exactly this standard practice.
>
> If we miss some essential things about this Gaussian policy, we hope that the reviewer can specify the concern.
>
> To make the setting explicit, we have added the following clarification at the beginning of Section 4:
>
> - *"The theoretical results in this section are based on the assumption of Gaussian action distributions, and are compatible with most mainstream off-policy RL methods."*
>
> &nbsp;
>
> **Q2. About Lemma 1, Lemma2, and the following theorem.**
>
> The intention of Section 4 is not to introduce technically heavy or deep CMDP theory, but to formally answer the research question: **how to control the safety cost of optimistic exploration during data collection.**
>
> - **Lemma 1** shows that the Policy-MGDA direction in the action space increases reward while not increasing cost.
>
> - **Lemma 2** then links this direction with a cost upper bound on the exploratory action, giving an explicit rule for choosing the step length so that the expected cost respects the threshold.
>
> Individually, these lemmas are elementary, but together they provide a principled and verifiable foundation for cost-constrained optimistic exploration, thus answering the question here. The theorem following these 2 lemmas is the same as the single-objective OAC (Eq. 6). Now we use Lemma 1 and 2 to derive the correct, cost-constraint compliant exploration gradient in the multi-objective safe RL task, so we can directly plug in the final gradient $u^*$ in (Eq. 6)
>
> To address the reviewer's concern, we make the connection between the OAC thereom (Eq. 6) and the two lemmas more explicit by adding the following explanations:
>
> - *"The exploration policy for data collection is given by the OAC theorem (Ciosek et al., 2019):"*
>
> - *"By using the two lemmas above, we can get the cost-constraint compliant exploration direction $u*$. Inserting it back into the OAC theorem in Eq. 6 gives the exploration policy of COX."*

---

> ### Author Response · Authors · 2025-11-22
> **Official Responses of Submission4892 to Reviewer ARdC - Part 2**
>
> **Q3. How do theoretical results contribute to improving sample efficiency or mitigating estimation bias? The empirical evaluations suggest some performance gains, it remains unclear under what circumstances the proposed method is effective. The framework does not appear to generalize well.**
>
> The reviewer appears confused about the connections between the proposed methods and the sample efficiency improvement / estimation bias mitigation, probably due to the unclear expression of our research question. Let us clarify the logic step by step:
>
> **(1) What is (and is not) our sample-efficiency claim?**
>
> Our starting point is the well-known challenge in constrained safe RL:
>
> - **On-policy methods:** typically satisfy cost constraints well in both data collection and deployment, but are sample-inefficient due to the on-policy data paradigm.
>
> - **Off-policy methods:** inherently have high sample efficiency, but they struggle with constraint satisfaction, because exploration is unconstrained and TD learning is sensitive to cost underestimation.
>
> Our goal is **not** to make off-policy RL more sample-efficient than it already is, but to **maintain** the high sample efficiency of off-policy RL while significantly improving **cost-constraint satisfaction** in both data collection and testing.** So, the contribution is to fix two key failure modes of off-policy safe RL, without giving up its inherent data-efficiency advantage over on-policy methods.
>
> This is now explicitly reflected in our research question (one word changed):
>
> - "How can we achieve off-policy safe RL that **maintains** high data efficiency and achieves robust constraint satisfaction in both data collection and deployment, through cost-constrained exploration and reliable value learning?"
>
> **(2) Role of Sections 4 and 5**
>
> Section 4 addresses the first failure mode: unsafe optimistic exploration during data collection. In the revised manuscript, we now state at the beginning of Section 4:
>
> - *"This section addresses the first challenge: cost-constrained exploration during data collection, while preserving the off-policy training pipeline and its sample-efficiency properties."*
>
> Section 5 tackles the second failure mode: underestimation bias in cost value learning under TD updates. We emphasize this at the beginning of Section 5:
>
> - *"This section mitigates the underestimation bias in cost estimation by using TQC and conservative value learning."*
>
> And at the end of Section 5, we add:
>
> - *"Combining COX and TQC conservative learning yields the full COX-Q algorithm. It addresses both cost-constrained exploration and underestimated cost in an integrated off-policy framework that maintains the inherent sample efficiency of off-policy RL methods."*
>
> **(3) What is the advantage of COX-Q?**
>
> Compared to on-policy baselines, COX-Q **maintains the expected off-policy sample-efficiency advantage** (learning with far fewer samples). Compared to off-policy baselines, COX-Q has similar sample efficiency but **better training and test-time safety costs**.
>
> We hope these explanations and modifications can address your concerns.
>
> &nbsp;
>
> **Q4. About the presentation and organization of the experimental section.**
>
> We thank the reviewer for carefully going through the experimental section. We hereby give our intended logic for the experimental section as a reference.
>
> At the beginning of Sec. 6, we explained that the selected 3 benchmarks represent three typical safety-critical regimes:
>
> - dense and immediate cost signals,
>
> - sparse reward and sparse cost,
>
> - zero-cost threshold.
>
> This is meant to show how COX-Q behaves under qualitatively different safety structures. Each subsection then motivates what is new compared to the previous benchmark. For instance, in the SMARTS autonomous driving experiments, we emphasize that:
>
> - *"The objects in the previous safe navigation tasks follow certain motion patterns or stay static. In the third experiment, we evaluate COX-Q in challenging autonomous driving tasks in which surrounding vehicles have closed-loop interactions with our RL agent."*
>
> Within each subsection, the explanation of results follows a consistent pattern. We first state the overall observation, e.g.,
>
> - *"Overall, COX-Q demonstrates high sample efficiency, achieves high cumulative returns, and has nearly-zero test costs after convergence, while keeping data collection costs below the predefined budget."*
>
> We then discuss specific advantages or disadvantages of each method and highlight key observations, e.g.,
>
> - *"CAL only applies conservative cost learning without exploration, therefore having the lowest data collection costs but also worse test performance than COX-Q.”*
>
> Could you please kindly specify, or provide some examples, about which parts or points are not logical or coherent? We would appreciate that and further improve the organization in the revised version.

---

> ### Author Response · Authors · 2025-11-22
> **Official Responses of Submission4892 to Reviewer ARdC - Part 3**
>
> **Q5. No quantitative results are provided to support the claim of reduced estimation bias.**
>
> Thank you for raising this point. Our intent here was not stated clearly enough. Our goal is **not to minimize** estimation bias, but to **mitigate underestimation** of the cost near the safety threshold. As pointed out by recent work [1,2], a slightly conservative (over-estimated) cost is often desirable in constrained RL, because it helps maintain constraint satisfaction. COX-Q uses TQC to make the cost critic conservative in this sense, not to make the bias close to zero.
>
> In constrained RL, the practically relevant quantity is not the raw Bellman error but the realized constrained cost (during data collection and at test time). These are quantitatively reported in our experiments:
>
> - **SafetyVelocity:** COX-Q achieves significantly lower test costs than other off-policy baselines, has well-controlled data-collection costs, while maintaining competitive or better returns. If the cost critic were strongly underestimated, this would manifest as frequent constraint violations in tests and high training costs (like ORAC), which is not what we see for COX-Q. Thus, the cost curves themselves are the quantitative evidence that underestimation is effectively mitigated in this regime.
>
> - **SafeNavigation:** Here, cost signals are sparse and underestimation is more severe. For this reason, we **do explicitly visualize the cost estimation bias** in the bottom row of Figure 2. The figure shows that early in training, underestimation happens with large training and testing cost violations simultaneously; as learning progresses, the bias is reduced, and the violations decrease accordingly. This is a direct quantitative link between cost underestimation and observed safety violations.
>
> To avoid confusion, we
>
> - (i) made the expression consistent to say “mitigating underestimation of the cost critic” throughout the paper;
>
> - (ii) explicitly pointed in the text (Sec. 5 and Sec. 6) to the training/test cost curves as quantitative evidence of underestimation bias;
>
> - (iii) explicitly pointed out that the bias visualization in Figure 2 is a direct illustration of how underestimation affects constraint violations.
>
> We hope that the explanations above can address your concern.
>
> [1] *Wu et al., Off-policy primal-dual safe reinforcement learning. In ICLR, 2024*
>
> [2] *Gao et al., Controlling Underestimation Bias in Constrained Reinforcement Learning for Safe Exploration (2025)*
>
> &nbsp;
>
> **Q6. There is no pseudo-code for clarification and implementation.**
>
> In our initial submission, the pseudo-code and implementation details of COX-Q are already provided in Appendix B. We did not include them in the main text due to the ICLR 9-page limit, and we explicitly mentioned this in the paper (lines 260–261):
>
> - *"The pseudo-code of COX-Q, the key differences from other baselines, and more details are provided in Appendix B."*
>
> We hope that the reviewer can comprehend this choice.
>
> &nbsp;
>
> Thank you again for your thoughtful feedback. We have revised the manuscript to better articulate the novelty and practical impact of our study. We hope these clarifications are persuasive and that you will consider a reassessment of our work.

---

### Official Review · Reviewer_EnSs · 2025-11-01

**Soundness:** 2
**Presentation:** 3
**Contribution:** 2
**Rating:** 6
**Confidence:** 4

**Summary:**

The paper proposes COX-Q, an off-policy safe reinforcement learning algorithm that achieves efficient and safety-aware exploration by integrating optimism, gradient conflict resolution, and uncertainty estimation. Building upon the optimistic actor-critic framework, COX-Q introduces a cost-constrained optimistic exploration strategy that balances reward maximization with constraint satisfaction. It employs a Policy-MGDA mechanism to align reward and cost gradients, ensuring exploration occurs only in directions that improve performance without increasing risk, and an adaptive step-length controller to prevent constraint violations during updates. Additionally, COX-Q leverages Truncated Quantile Critics (TQC) to obtain uncertainty-aware and bias-reduced value estimates.

**Strengths:**

1. The paper tackles an important and timely problem by addressing the challenges associated with implementing off-policy reinforcement learning algorithms in the context of safe RL, where maintaining safety during exploration remains a key concern.

2. The proposed framework integrates gradient conflict resolution with mechanisms to mitigate value estimation biases, effectively addressing both reward overestimation and cost underestimation—two critical issues that commonly affect off-policy safe RL methods.

3. The paper presents a comprehensive empirical evaluation across diverse domains, including locomotion, navigation, and autonomous driving tasks, and benchmarks the proposed method against strong and relevant baselines.

**Weaknesses:**

[W1] Relevant Comparisons are Missing

The concept of gradient manipulation or resolution is not entirely new in Safe RL, as several works [1–3] have proposed similar approaches. To convincingly demonstrate the advantages of the gradient resolution method presented in this paper, it is essential to compare against these baselines. While some of these methods report results in the context of on-policy RL, their underlying ideas can be readily adapted to off-policy settings.

Recent work [4] addresses the problem of cost underestimation, which is also a focus of our method. Therefore, it represents an important baseline for evaluating the effectiveness of our approach in mitigating cost underestimation.

[W2] Lack of Ablations

The proposed method comprises two key components: (i) cost-constrained optimistic exploration via Gradient Resolution, and (ii) mitigation of value estimation bias via Truncated Quantile Critics (TQC). The paper would benefit significantly from ablation studies that quantify the contribution of each component individually. Specifically, it would be valuable to demonstrate:

1. the performance of gradient resolution alone compared to prior works [1–3], and

2. the effect of addressing value estimation bias via TQC compared to [4].

Such ablations would clearly establish the incremental value of each component and provide a stronger empirical justification for the proposed approach.

References
[1] Gu et al., Balance Reward and Safety Optimization for Safe Reinforcement Learning: A Perspective of Gradient Manipulation (2024)
[2] Chow et al., Safe Policy Learning for Continuous Control (2020)
[3] Liu et al., Constrained Variational Policy Optimization for Safe Reinforcement Learning (2022)
[4] Gao et al., Controlling Underestimation Bias in Constrained Reinforcement Learning for Safe Exploration (2025)

**Questions:**

My questions pertain to the weaknesses highlighted above:

1. How does the proposed gradient resolution method for optimistic exploration differ from existing approaches in the literature?

2. In the context of safe exploration, preventing constraint violations during early learning is critical. While overestimation of cost can promote safe exploration, it may also lead to overly conservative behavior. How does the proposed method better balance safety and exploration compared to existing baselines such as MICE [1]?

Reference
[1] Gao et al., Controlling Underestimation Bias in Constrained Reinforcement Learning for Safe Exploration (2025)

---

> ### Author Response · Authors · 2025-11-22
> **Official Responses of Submission4892 to Reviewer EnSs - Part 1**
>
> We thank the reviewer for the careful reading of our manuscript and the constructive feedback. According to the content, the raised questions are categorized and reorganized for coherence and clarity. Our responses are provided below.
>
> &nbsp;
>
> **Q1.1 Relevant Comparisons are Missing. [1–3] have proposed similar approaches. While some of these methods report results in the context of on-policy RL, their underlying ideas can be readily adapted to off-policy settings.**
>
> **Q1.2 How does the proposed gradient resolution method for optimistic exploration differ from existing approaches?**
>
> **Q1.3 Demonstrate the performance of gradient resolution alone compared to prior works [1–3].**
>
> The first group of questions mainly concerns the novelty and positioning of our gradient conflict resolution method (Policy-MGDA) relative to existing gradient manipulation methods [1–3].
>
> We thank the reviewer for pointing us to these highly relevant works. We did carefully study [1–3] when preparing our submission. However, our **Policy-MGDA plays a fundamentally different role in the RL pipeline and is not directly comparable with these baselines.** We explain this in more detail below:
>
> - **(1) Different stage in the training:** The gradient manipulations in [1–3] are all applied in the offline update stage, typically during the actor update. In contrast, Policy-MGDA operates in the online data collection stage, where both the actor and critic parameters are frozen. We only manipulate the exploration action at each state, based on the critics. Thus, [1–3] aim to modify how the policy is trained, whereas our method modifies how data is collected from the environment.
>
> - **(2) Different space for gradient manipulation:** In [1–3], gradients are manipulated in the space of policy network parameters, whose dimensionality can be very large. Policy-MGDA instead manipulates gradients with respect to the action, which is typically of much lower dimensionality. This makes our approach conceptually and practically different. To the best of our knowledge, the gradient conflict in the action space in exploration has not been addressed in the literature yet. This is one of our core contributions.
>
> - **(3) Different RL paradigm:** As the reviewer notes, [1–3] are designed for **on-policy primal** constrained RL methods. Extending these approaches to the off-policy setting is highly non-trivial: distributional shift makes it difficult to construct reliable safety sets or barriers when using replay buffers. To the best of our knowledge, there are currently no successful off-the-shelf applications of such primal methods to off-policy deep safe RL.
>
> - **(4) The primal-dual approach does not need gradient manipulation in the actor updates:** Our COX-Q and all baselines use the primal-dual approach, which is theoretically the correct way: gradient conflict is the nature of the constraint--the algorithm is supposed to trade off reward vs cost via the Lagrangian multiplier $\lambda$. There is no requirement to "orthogonalize" or "project" gradients like in MGDA/PCGrad, because we don’t want to treat reward and cost as symmetric objectives; cost is a constraint (hard priority), not just "another objective" like in the primal approach.
>
> Taken together, these points explain why adapting [1–3] as baselines in our off-policy primal–dual setting would **not be beneficial**, and why Policy-MGDA addresses a different part of the problem (safe exploration) rather than competing directly with parameter-space gradient manipulation in primal on-policy methods.
>
> To better avoid confusion and to clarify the novelty of Policy-MGDA, we have added the following sentence at the end of Sec. 4.1:
>
> - *"Note that Policy-MGDA operates in the action space during the online data collection stage with frozen network parameters, which makes it fundamentally different in both role and design from existing gradient manipulation methods [1–3]."*
>
> &nbsp;

---

> ### Author Response · Authors · 2025-11-22
> **Official Responses of Submission4892 to Reviewer EnSs - Part 2**
>
> **Q2. Compare the proposed approach against MICE [4] w.r.t cost underestimation mitigation**
>
> We thank the reviewer for referring to this relevant ICML 2025 paper, which we had not yet identified when preparing our ICLR submission.
>
> We have studied MICE carefully and attempted to reproduce the method. MICE was implemented in two on-policy safe RL methods in [4]. All baselines are also on-policy. The reviewer asked to compare MICE with COX-Q. Therefore, they should be integrated in the same off-policy framework. However, we tried but failed to put MICE layer in an primal-dual off-policy framework.
>
> MICE has three pieces that are agnostic to on- vs off-policy: (1) memory of unsafe states (embedded + kNN)，(2) intrinsic cost as pseudo-count of similarity to unsafe states, and (3) an extrinsic–intrinsic cost critic. But, the following components are not compatible with off-policy RL:
>
> - **(1) Worst-case constraint violation bound:** Off-policy has no explicit KL trust-region. Due to the data replay + distributional shift, the theorem doesn’t directly apply to off-policy methods.
>
> - **(2) Value convergence proof:** Their convergence result for the extrinsic–intrinsic cost value function assumes on-policy TD under standard Robbins–Monro conditions. Off-policy deep SAC critics are already outside that theory; adding intrinsic cost doesn’t fix that.
>
> - **(3) Clean separation between current policy and memory data:** In [4], memory stores unsafe states from recent rollouts of the policy, which matches the trust-region assumption. In off-policy approaches, replay is mixed over many past policies. We have a fuzzier notion of "unsafe under current policy".
>
> Due to these reasons, we still cannot successfully train the off-policy RL agents after adding the MICE layer. At this stage, our conclusion is that **MICE is not directly applicable to the off-policy approaches.** If the reviewer has any ideas about how to combine them, we would be happy to discuss and try in the remaining time of the discussion period.
>
> We do agree that this paper tries to address a similar concern as ours. Therefore, we add it to the literature review section:
>
> - *"Gao et al. proposed MICE to address the underestimation of cost in the on-policy method. The key idea is to use a memory-based intrinsic cost around unsafe states so the cost critic conservatively overestimates risk."*
>
> &nbsp;
>
> **Q3. Lack of ablations of the contribution of each component individually.**
>
> We agree that ablations are important. On SafetyVelocity, our original Figure 1 already contained most of the needed ablations, but the naming obscured the structure, and a pure TQC baseline was missing. In fact:
>
> - **ORAC = TQC + vanilla OAC**
>
> - **ORAC (with step length auto-tuning) = TQC + OAC + step length adjustment, no gradient conflict resolution.**
>
> - **COX-Q = TQC + OAC + step length adjustment + gradient conflict resolution**
>
> We acknowledge that the name ORAC was confusing. Shortly before submission, we found the ORAC preprint [5], which uses IQN (a quantile critic similar to TQC) + vanilla OAC. Since no code or SafetyVelocity hyperparameters were available, we implemented a TQC-based version and kept the ORAC name.
>
> We accept the suggestion of the reviewer and have now (i) **renamed the baselines to this explicit form** and (ii) **added the missing TQC-only baseline.** The results in Figure 1 were updated in the new version.
> The updated figure shows:
>
> - **TQC** improves **sample efficiency** over point-estimate critics.
>
> - **TQC + OAC** improves **test-time cost**(lower) compared to TQC alone.
>
> - **Step-length auto-tuning** keeps **training (data-collection) cost below the threshold after convergence.**
>
> - **Gradient conflict resolution** further improves **training cost control in the early–middle phase.**
>
> Thus, each component has a clear effect on SafetyVelocity. For SafeNavigation and SMARTS, the training time is substantially longer, so we focus on detailed multi-seed ablations on SafetyVelocity, where they are computationally easier.
>
> We will upload the revised version later, as other reviewers may request reorganization of some sections. This response is for timely discussion. We will notify the reviewer once the updated PDF is available.
>
> [5] *James McCarthy et al., Optimistic exploration for risk-averse constrained reinforcement learning. arXiv preprint arXiv:2507.08793, 2025.*

---

> ### Author Response · Authors · 2025-11-22
> **Official Responses of Submission4892 to Reviewer EnSs - Part 3**
>
> **Q4. How does COX-Q balance safety and exploration compared to existing baselines such as MICE?**
>
> This is a very insightful comment. For the current COX-Q paper, we would like to be transparent: **we do not claim a clear advantage over all existing methods, specifically in terms of optimally balancing safety and exploration during early learning.**
>
> In our current design, reward and cost are modelled by two **independent** ensembles of critics, and TQC is applied separately to reward and cost. As a result, the main control we have over the safety–exploration trade-off is through how many quantile atoms are truncated in the reward and cost critics, respectively. While this allows us to adjust conservatism vs. exploration in practice, it requires careful hyperparameter tuning and does not yet constitute a fully principled mechanism for automatically balancing the two.
>
> We view this as an important limitation. A plausible solution is to explicitly **model the joint distribution and correlation between reward and cost**, and then exploit this structure to control the safety–exploration trade-off via the chain rule. This provides a more systematic way to prioritize safety while still encouraging optimistic exploration in directions that are consistent with a conservative view of cost. This conceptual picture reflects the direction we are pursuing to better address the concern raised by the reviewer.
>
> &nbsp;
>
> We hope that our detailed responses can address the reviewer's concerns and help evaluate the contribution of our study. If the reviewer has any further questions, we would be happy to discuss them.

---

### Official Review · Reviewer_cj7m · 2025-11-01

**Soundness:** 3
**Presentation:** 3
**Contribution:** 2
**Rating:** 4
**Confidence:** 2

**Summary:**

The paper tackles a key gap in off-policy safe RL: optimistic exploration improves sample efficiency but often violates cost constraints, because reward and cost gradients point in different directions and cost critics are biased. The authors propose a primal-dual off-policy algorithm that performs cost-constrained optimistic exploration, and adaptively shrinks the exploration step.

**Strengths:**

1. The paper clearly identifies the off-policy safety pain point, i.e., optimism without cost control, and a concrete mechanism of policy-MGDA + cost-aware step-length selection to make OAC-style exploration safe.
2. Both key steps are derived with closed-form solutions. The SMARTS results, though single-seeded, are non-trivial and show the method still works with large networks.

**Weaknesses:**

1. Safety still depends on cost-critic accuracy. The whole cost-bounded exploration, c.f. Sec. 4.2., assumes the lower/mean cost estimates are reliable. In sparse-cost navigation the paper shows underestimation, and COX-Q behaves like ORAC. A discussion for robustifying the cost critic is missing. I am curious how the authors would further improve this.
2. The lemmas give local, per-step solutions, but there is no theorem that the whole off-policy procedure respects the CMDP constraint under function approximation and replay. However the abstract emphasizes controlled data-collection cost. I believe this gap should be made explicit.
3. Some baselines are slightly weakened. CAL is run with UTD=1 but CAL's strength is precisely high UTD.

**Questions:**

Please see weaknesses.

---

> ### Author Response · Authors · 2025-11-22
> **Official Responses of Submission4892 to Reviewer cj7m - Part 1**
>
> We thank the reviewer for giving insightful comments. The major concerns are about the accuracy of the cost-critic learning and how it impacts the constraint compliance during the entire training procedure. Our responses are given as follows.
>
> &nbsp;
>
> **Q1. About the cost-critic accuracy.**
>
> This is a very insightful and critical point. As our results show, for sparse cost tasks like SafeNavigation, the main bottleneck is not the exploration mechanism but the accuracy of value learning. In other words, **safety still fundamentally hinges on the quality of the cost-critic.** We agree that there should be an explicit discussion on how to robustify the cost-critic learning.
>
> We added the following paragraph in Section 6.2:
>
> - *"The result indicates that, for constrained RL with sparse costs, the estimation bias in the cumulative cost is the major bottleneck, rather than the exploration mechanism. For off-policy approaches, the cost learning can be made more robust by, e.g., using multi-step returns / TD learning, prioritized experience replay [1], or Hindsight Experience Replay (HER) [2]."*
>
> In the final limitation discussion, we also added a future research direction:
>
> - *"Another future research direction is how to effectively implement COX in sparse-cost tasks. A plausible solution is to use, e.g., HER [2] or prioritized experience replay [1] to robustify the cost-critic learning."*
>
> [1] *Schaul, Tom, et al. "Prioritized experience replay." ICLR 2016*
>
> [2] *Andrychowicz, Marcin, et al. "Hindsight experience replay." NeurIPS/NIPS 2017*
>
> &nbsp;
>
> **Q2. The lemmas give local, per-step solutions, but there is no theorem that the whole off-policy procedure respects the CMDP constraint under function approximation and replay. However, the abstract emphasizes controlled data-collection cost. This gap should be made explicit.**
>
> This question actually contains two points. We clarify them separately.
>
> **(1) Local solution:** Although the solution appears local and per-step, the underlying computation is based on the distribution of discounted cumulative reward and cost. Therefore, when making each per-step decision, **the agent is already taking the future rewards and costs into account through the learned return distributions**, rather than optimizing an immediate, myopic quantity.
>
> **(2) Approximation errors:** If the "whole off-policy procedure" here refers to the entire training process (we assume this is the intended meaning), then the reviewer’s comment is entirely valid. With function approximation using neural networks, experience replay, and TD learning, we cannot guarantee that the critic (especially the cost critic) always provides accurate estimates. This issue is particularly pronounced in the early stage of training. For example, in both SafetyVelocity and SafeNavigation benchmarks, our methods exhibit unstable, fluctuating, or spiking training costs at the beginning of learning. We view this as an inherent limitation of deep model-free RL, rather than a specific flaw of COX.
>
> A possible remedy is to combine the proposed COX mechanism with hard safety-constrained methods, such as Control Barrier Functions (CBFs) and reachability-based approaches [1], especially in the early stage of training. Alternatively, implementing COX within model-based safe RL methods, e.g., SafetyDreamer [2], is expected to mitigate this issue by leveraging a learned world model. These are promising future research directions.
>
> To address this concern, we make this gap explicit at the end of Sec. 4:
>
> - *"So far, we have explained the ‘COX-’ part, including the effective exploration direction and the adaptive step length under cost constraints. It is useful to note that the theories in this section are based on accurate return estimation. If the critics cannot provide reliable return distributions due to a lack of data or function approximation errors, especially in the early stage of training, the data-collection cost cannot be effectively controlled. Potential improvements include incorporating classical methods such as reachability analysis [1], or combining COX with model-based RL, such as SafetyDreamer [2]."*
>
> [1] *Milan Ganai, Zheng Gong, Chenning Yu, Sylvia Herbert, and Sicun Gao. Iterative reachability estimation for safe reinforcement learning. Advances in Neural Information Processing Systems, 36:69764–69797, 2023.*
>
> [2] *Weidong Huang, Jiaming Ji, Chunhe Xia, Borong Zhang, and Yaodong Yang. Safedreamer: Safe reinforcement learning with world models. arXiv preprint arXiv:2307.07176, 2023.*

---

> ### Author Response · Authors · 2025-11-22
> **Official Responses of Submission4892 to Reviewer cj7m - Part 2**
>
> **Q3. Use UTD=1 for CAL**
>
> We agree that CAL’s original implementation uses a high UTD ratio. However, in our experiments, we deliberately set UTD=1 for CAL, not to weaken it, but **to isolate and fairly compare the effect of conservative distributional value learning and the proposed COX exploration mechanism, without confounding from aggressive critic updates.**
>
> Concretely, CAL combines 3 components:
>
> - Augmented Lagrangian Method (ALM) – the core contribution.
>
> - Conservative cost Q-value estimation.
>
> - Ensembled Q-learning with high UTD, which is mainly responsible for its strong sample efficiency in the original paper.
>
> It is well known that using ensembles with a high UTD ratio (e.g., REDQ [1], MaxMin Q-learning [2]) can significantly boost sample efficiency for many methods, not only CAL.
>
> In our work, COX-Q also uses ALM, replaces CAL’s conservative cost Q-learning by distributional critics, and introduces the COX exploration mechanism. Our goal in this paper is to compare “conservative/distributional critics + COX exploration” against other constrained RL methods under a common, standard UTD setting (UTD=1), rather than to compare which algorithm benefits more from high-UTD ensembles. If we were to run CAL with UTD=20 (its original high-UTD setting), to be methodologically fair, we would also need to equip all baselines with similarly strong ensembles and high UTD. This would largely turn the comparison into a study of ensembled Q-learning + high UTD, which is orthogonal to our main focus and would obscure the specific contribution of COX and distributional critics.
>
> We acknowledge that this design choice was not clearly explained in the original submission. To clarify this in the revised manuscript, we added the following sentence when introducing the baselines:
>
> - *"We choose UTD=1 for CAL so that the impact of high UTD ratios is excluded. Using UTD=1 for all baselines makes the role of conservative (or distributional) cost learning and the proposed exploration mechanism comparable across methods."*
>
> For other baselines, we either use the released official code, the implementation in the OmniSafe platform, or our own implementation following the original paper (e.g., ORAC).
>
> [1] *Chen, Xinyue, et al. "Randomized ensembled double q-learning: Learning fast without a model." ICLR 2021*
>
> [2] *Lan, Qingfeng, et al. "Maxmin q-learning: Controlling the estimation bias of q-learning." arXiv preprint arXiv:2002.06487 (2020).*
>
> &nbsp;
>
> We hope that our responses adequately address your concerns. If any points remain unclear or if you have additional questions, we would be very happy to further clarify and discuss.

---

### Author Response · Authors · 2025-11-22
**Official comment of the 1st-round discussion**

Dear all,

The authors appreciate the effort the reviewers put into reviewing our paper and giving insightful and constructive feedback, which is important for improving our study. We have posted the point-by-point responses to the raised questions and revised our paper accordingly. In the uploaded revised manuscript, all modifications are marked by blue text or in blue boxes. Particularly:

- Abstract, research question, and some key expressions (e.g., "underestimation bias mitigation") were adjusted based on the reviewers' comments.

- The experimental section was reorganized for clarity.

- Figure 1 was updated for the ablation study.

We hope that these revisions and responses can address the reviewers' concerns. If the reviewers have any further questions, we would be happy to discuss them.

---

### Author Response · Authors · 2025-12-02
**A Factual Summary of the Rebuttal**

To facilitate the evaluation of our study, we provide a high-level **factual** summary of the rebuttal for AC and all readers.

&nbsp;

**Main concerns:**

**1. Positioning of the paper:** (1) Relation of Policy-MGDA to prior gradient manipulation methods. (2) Comparison and connection to recent cost-underestimation work, e.g., MICE.

**2. Theory clarity:** (1) The Gaussian policy assumption. (2) Lemmas 1 & 2 are not connected to a following stronger theorem. (3) No global guarantee that the CMDP constraint is always satisfied under function approximation.

**3. Limitations on sparse costs:** (1) Safety depends on cost-critic accuracy in sparse-cost tasks. (2) No quantitative support for reduced estimation bias.

**4. Experiments:** (1) Lack of ablation studies. (2) Experimental section perceived as hard to follow. (3) One reviewer claimed no pseudo-code.

&nbsp;

**Main clarifications and revisions:**

**1. Positioning vs prior work**

- Clarified that the proposed Policy-MGDA acts in action space during online data collection with frozen networks. It is fundamentally different from manipulating policy parameters during offline updates.

- Analyzed MICE and found that it is not compatible with the off-policy safe RL framework.

**2. Assumptions and theory**

- Explained that the Gaussian policy is common in RL.

- Clarified that Lemma 1 (direction) + Lemma 2 (step length) together give a principled construction of cost-constrained optimistic exploration. The following theorem has already been given in a prior optimistic actor-critic (OAC) paper.

- Explicitly stated the limitations of no global guarantees.

**3. Limitations on sparse costs**

- Explicitly stated the limitations on sparse-cost tasks.

- Clarified that the goal is mitigating underestimation of cost near the constraint boundary, not eliminating bias; Pointed to existing training/test cost curves (SafetyVelocity) and explicit bias visualization (SafeNavigation, Fig. 2 bottom row) as quantitative evidence.

**4. Ablations and experiments**

- Added ablations in updated Fig. 1.

- Re-organized the experiment and discussion section for clarification.

- Pointed out that pseudo-code is already in Appendix B, as referenced in the main text.

&nbsp;

**Scores:**

No reviewers responded before the rebuttal was locked.

&nbsp;

We hope that this factual summary is helpful for better understanding our work and the rebuttal.

---

### Meta-Review · Area_Chair_sQhQ · 2026-01-07

**Summary:**

**Summary of the paper**
This paper proposes COX-Q, an off-policy safe reinforcement learning algorithm designed to enable optimistic exploration under cost constraints. The method integrates three key components: (i) Policy-MGDA, which resolves reward–cost gradient conflicts during online data collection by operating in the action space; (ii) an adaptive step-length controller that limits exploration steps to prevent constraint violations; and (iii) distributional critics (TQC) to mitigate reward overestimation and cost underestimation. COX-Q is implemented within a primal–dual framework and evaluated across locomotion, navigation, and autonomous driving benchmarks, including SMARTS. Empirical results demonstrate that COX-Q achieves competitive performance while improving safety during data collection.

**Summary of the reviewers' concern**

Reviewer cj7m rated the paper as marginally below acceptance and identified several concerns. The reviewer emphasized that safety ultimately depends on the accuracy of the learned cost critic, particularly in sparse-cost environments, where underestimation can cause COX-Q to behave similarly to unconstrained optimistic methods. They also noted a theoretical gap: while the paper derives per-step closed-form solutions for safe exploration, it does not provide a global guarantee that the full off-policy training procedure satisfies CMDP constraints under function approximation and replay. Additionally, this reviewer expressed concern that some baselines, specifically CAL, were evaluated under weaker settings (UTD = 1), potentially understating their performance.

Reviewer EnSs rated the paper as marginally above acceptance, but raised concerns about comparative evaluation and ablations. This reviewer noted that gradient manipulation ideas have appeared previously in safe RL and requested clearer differentiation from prior gradient-based methods, particularly those operating in the policy-parameter space. Reviewer also asked for comparisons with recent work addressing cost underestimation (e.g., MICE) and for clearer ablation studies isolating the contributions of Policy-MGDA and TQC.

**Reviewer Concerns:**

The rebuttal substantively addresses the core concerns raised by both reviewers.

In response to Reviewer cj7m, the authors explicitly acknowledge that cost-critic accuracy is a fundamental bottleneck and add a clear discussion of this limitation, along with concrete directions for improving robustness (e.g., multi-step TD learning, prioritized replay, HER). They also clarify that global safety guarantees are not claimed, and revise the manuscript to explicitly state that accurate return estimation is a prerequisite for controlled data-collection cost. Regarding baseline fairness, the authors provide a detailed and methodologically sound justification for using UTD = 1 for CAL, explaining that their goal is to isolate the effect of conservative/distributional critics and the proposed COX exploration mechanism, rather than conflating results with high-UTD ensemble effects.

In response to Reviewer EnSs, the rebuttal carefully distinguishes Policy-MGDA from prior gradient manipulation methods by emphasizing differences in training stage (online data collection vs. offline updates), space of operation (action space vs. parameter space), and RL paradigm (off-policy primal–dual vs. on-policy primal methods). The authors also explain why recent methods such as MICE are not directly applicable to off-policy primal–dual settings and add them to the related work for completeness. Finally, the authors clarify and strengthen the ablation results by explicitly decomposing ORAC, TQC, step-length tuning, and gradient conflict resolution, showing the contribution of each component.

**Reviewer Scores:**

Reviewer cj7m is very likely to increase their score from 4 to 6, as almost all concerns raised by this reviewer are clearly addressed in the rebuttal. Reviewer EnSs is most likely to keep his/her 6/10 score, and may even increase to 8/10.

---

### Decision · Program_Chairs · 2026-01-26

Accept (Poster)